# MRTF-dependent cytoskeletal dynamics drive efficient cell cycle progression

Julie C. Nielsen*, Maria Benito-Jardon, Noel Christo Petrela, Jessica Diring‡, Sofie Bellamy§ and Richard Treisman¶

## ABSTRACT

Serum response factor (SRF) and its cofactors, myocardin-related transcription factors A and B (MRTF-A and MRTF-B, respectively), regulate transcription of numerous cytoskeletal structural and regulatory genes, and most MRTF/SRF inactivation phenotypes reflect deficits in cytoskeletal dynamics. We show that MRTF–SRF activity is required for effective proliferation of both primary and immortalised fibroblast and epithelial cells. Cells lacking the MRTFs or SRF proliferate very slowly, express elevated levels of senescence-associated secretory phenotype (SASP) factors and senescence-associated β-galactosidase activity, and inhibit proliferation of co-cultured primary wild-type cells. They exhibit decreased levels of CDK1 and CKS2 proteins, and elevated levels of CDK inhibitors, usually p27 (also known as CDKN1B). These phenotypes, which can be fully reversed by re-expression of MRTF-A, are also seen in wild-type cells arrested by serum deprivation. Moreover, in wild-type cells direct interference with cytoskeletal dynamics through inhibition of Rho kinases (ROCKs) or myosin ATPase induces a similar proliferative defect to that seen in MRTF-null cells. MRTF-null cells exhibit multiple cytoskeletal defects and markedly reduced contractility. We propose that MRTF–SRF signalling will be required for cell proliferation in cell types and environments where physical progression through cell cycle transitions requires high contractility.

KEY WORDS: MRTF-A, MRTF-B, SRF, MKL1, Cytoskeleton, Cell cycle, Quiescence, Senescence

## INTRODUCTION

The cell cycle is controlled by a conserved network of cyclin–CDKs and their regulators that ensures that cell growth, genome replication and segregation, and cell division occur in the right order and at the right time (Malumbres and Barbacid, 2001; Morgan, 1997). In multicellular organisms, however, many cell types remain quiescent unless prompted to proliferate by external cues (Marescal and Cheeseman, 2020). Proliferation of cultured mammalian cells is thus dependent on external stimuli, including serum mitogens, nutrients and substrate adhesion, whose withdrawal or downregulation by other cues, such as cell confluence and matrix compliance, leads to cell cycle exit and return to quiescence (Otsuka and Moskowitz, 1975; Pardee, 1974; Pardee et al., 1978). These requirements and controls are relaxed in cultured tumour cells, suggesting that their disruption is important for oncogenic transformation (Malumbres and Barbacid, 2001). Stress stimuli such as telomere erosion or oncogene activation lead to senescence, a distinct non-proliferative state considered irreversible (Campisi and d'Adda di Fagagna, 2007; Salama et al., 2014). Senescent cells secrete factors that promote cell cycle arrest in both an autocrine and paracrine manner (Acosta et al., 2013; Coppe et al., 2008).

In non-transformed cells, the Ras–ERK and Rho GTPase signalling pathways are central to control of activity of the core cell cycle machinery by mitogenic stimuli (Pruitt and Der, 2001; Olson et al., 1995). Ras–ERK signalling is linked to expression of cyclin D mRNA and protein synthesis (Matsushime et al., 1991; Won et al., 1992), whereas Rho signalling controls focal adhesion assembly and signalling, and promotes actin-dependent cell contractility (Hotchin and Hall, 1995; reviewed by Burridge and Chrzanowska-Wodnicka, 1996). Proliferation requires Rho-dependent adhesion and cell spreading (Chen et al., 1997; reviewed by Kamranvar et al., 2022; Mammoto and Ingber, 2009), and Rho-regulated cytoskeletal dynamics play critical roles in mitosis and cytokinesis (Basant and Glotzer, 2018; Ramkumar and Baum, 2016). Classical studies showed that cell cycle re-entry from quiescence requires new RNA synthesis by pre-existing cellular factors (Pledger et al., 1981; Smith and Stiles, 1981). Mitogen-induced cellular immediate-early (IE) genes include c-myc (*Myc*) and c-fos (*Fos*), other transcription factors and actins (Cochran et al., 1983, 1984; Greenberg and Ziff, 1984; Kelly et al., 1983), and IE genes are also induced upon cell–extracellular matrix (ECM) adhesion (Dike and Farmer, 1988). Nevertheless, whether and how IE gene induction controls activation of cyclin D remains unclear.

Many IE genes are controlled by the SRF transcription factor (Norman et al., 1988; Schratt et al., 2001; Treisman, 1986). SRF works in partnership with two families of signal-regulated cofactors, the TCFs and the MRTFs, which respectively couple its activity to the Ras–ERK and Rho signalling pathways (Olson and Nordheim, 2010; Posern and Treisman, 2006), MRTF–SRF target genes include actins and many other cytoskeletal structural and regulatory proteins (Esnault et al., 2014; Olson and Nordheim, 2010), and cells lacking SRF or the MRTFs exhibit deficits in adhesion, motility and invasion (Alberti et al., 2005; Medjkane et al., 2009; Mokalled et al., 2010; Schratt et al., 2002). SRF-null mouse embryonic stem cells proliferate normally, as do embryos up to embryonic day (E)6 (Arsenian et al., 1998; Schratt et al., 2001), and cell types such as neuroblasts (Alberti et al., 2005; Medjkane et al., 2009; Mokalled et al., 2010; Schratt et al., 2002), pre-DP thymocytes (Mylona et al., 2011) and

Signalling and transcription Laboratory, Francis Crick Institute, 1 Midland Road, London NW1 1AT, UK.
*Present address: Department of Biology, Stanford University, Stanford, CA 94305-5020, USA. ‡Present address: Friedrich Miescher Institute for Biomedical Research, Fabrikstrasse 24, 4056 Basel, Switzerland. §Present address: Holmen Board and Paper, Iggesunds Bruk, 825 80 Iggesund, Sweden.

¶Author for correspondence (Richard.Treisman@crick.ac.uk)

J.C.N., 0000-0002-3803-0470; M.B.-J., 0000-0001-9562-5430; N.C.P., 0009-0000-1622-1455; J.D., 0000-0003-2348-3941; S.B., 0009-0000-9537-554X; R.T., 0000-0002-9658-0067

TCR-activated T cells (Maurice et al., 2024). In contrast, SRF inactivation induces apparent senescence in smooth muscle cells (Angstenberger et al., 2007; Werth et al., 2010) and activates oncogene-induced senescence in cancer cells lacking the DLC1 tumour suppressor, a RhoGAP (Hampl et al., 2013; Hermanns et al., 2017; Muehlich et al., 2012).

Here we revisit the issue of Rho-dependent cytoskeletal dynamics, MRTF–SRF activity and cell proliferation. We show that primary and immortalised fibroblasts and epithelial cells exhibit a proliferation defect with features of senescence and quiescence, but which can be reversed by MRTF-A re-expression. A similar proliferation defect is induced in wild-type (WT) mouse embryonic fibroblasts (MEFs) upon serum starvation or stringent inhibition of Rho kinases (ROCKs) or myosin ATPase, and MRTF-null cells exhibit cytoskeletal defects and reduced contractility. Our results support a model in which MRTF–SRF-linked cytoskeletal dynamics, particularly in cell adhesion and contractility, are essential for generation of the proliferative signal provided by cell adhesion.

## RESULTS
### The MRTFs are required for proliferation of SV40-immortalised MEFs
To investigate the role of the MRTFs in cell proliferation, we established pools of SV40-immortalised MEFs from animals expressing tamoxifen-inducible Cre in either WT ($Rosa26^{Tam-Cre}$) or conditional $Mrtf$-null ($Mrtfa^{-/-}$;$Mrtfb^{fl/fl}$;$Rosa26^{Tam-Cre}$) or conditional $Srf$-null ($Srf^{fl/fl}$;$Rosa26^{Tam-Cre}$) backgrounds. All the MEF pools grew at comparable rates. To induce gene inactivation, cells were treated with 4-hydroxytamoxifen (4OHT) and then plated for analysis 7 or 10 days later (Fig. 1A). Unless otherwise stated, we will hereafter refer to 4OHT-treated $Mrtfa^{-/-}$;$Mrtfb^{fl/fl}$;$Rosa26^{Tam-Cre}$ or $Srf^{fl/fl}$; $Rosa26^{Tam-Cre}$ cells as $Mrtfab^{-/-}$ and $Srf^{-/-}$ MEFs, respectively.

Untreated immortalised $Mrtfa^{-/-}$;$Mrtfb^{fl/fl}$;$Rosa26^{Tam-Cre}$ MEFs grew at similar rates to WT MEFs, indicating that MRTF-A is dispensable for growth. However, following 4OHT treatment all three pools of $Mrtfa^{-/-}$;$Mrtfb^{fl/fl}$;$Rosa26^{Tam-Cre}$ MEFs exhibited a severe proliferation defect compared to the WT MEF pools (Fig. 1B). At day 12, bromodeoxyuridine (BrdU) labelling was significantly reduced, whereas proportions of both G1 and G2/M cells increased, the proportion of apoptotic cells remaining unchanged (Fig. 1C–E). The $Mrtfab^{-/-}$ cells also had elevated levels of neutral β-galactosidase (senescence-associated β-galactosidase, SA-βGal) activity, conventionally regarded as a senescence marker (Dimri et al., 1995) (Fig. 1F). $Mrtfab^{-/-}$ MEFs exhibited elevated expression of senescence-associated secretory phenotype (SASP) factors (Acosta et al., 2013), including secreted proteins ($Il6$, $Mmp2$, $Hgf$), cellular receptors ($Tgfb2$, $Tgfb3$, $Tgfbi$, $Tlr2$) and downstream effectors ($Btg2$, $Irak1$, $Irak2$) (Fig. 1G, Fig. S1A). The SASP can act in a paracrine fashion to induce senescence (Acosta et al., 2013). We cultured $Mrtfab^{-/-}$ or WT cells with mCherry-expressing primary kidney fibroblasts for 5 days and assessed senescence by staining with CellEvent Senescence Green (CESG). Co-culture with $Mrtfab^{-/-}$ MEFs, but not WT MEFs, increased the proportion of senescent CESG-positive primary kidney fibroblasts, indicating that $Mrtfab^{-/-}$ cells can act in a paracrine manner to influence fibroblast proliferation (Fig. 1H).

Replicative senescence is generally associated with upregulation of p53 (also known as TP53) and the cyclin–CDK inhibitors (CKIs) p21 (CDKN1A) and p16 (CDKN2A), which inhibit the pro-proliferative CDK4/6–Rb–E2F pathway (Campisi, 2013). We therefore conducted immunoblot analysis of expression of these and other cell cycle markers in $Mrtfab^{-/-}$ MEFs (Fig. 1I). Each

$Mrtfab^{-/-}$ MEF pool displayed upregulation of p27, whereas p21 and cyclin D1 were downregulated, as were p53, CDK1 and CKS2, and p16 also decreased slightly (Fig. 1I; Fig. S1A). Previous studies have shown that p27 upregulation is associated with reversible cell cycle exit and quiescence (Chen et al., 1997; Coats et al., 1996; Huang et al., 1998; Klein et al., 2009; Polyak et al., 1994). In contrast, without 4OHT treatment, $Mrtfa^{-/-}$;$Mrtfb^{fl/fl}$ MEFs displayed a similar marker profile to WT cells (Fig. S1B). Moreover, while MRTF–SRF signalling has been shown to suppress oncogene-induced senescence and ERK activity in $DLC1$-deleted cancer cells (Hampl et al., 2013), our $Mrtfab^{-/-}$ MEFs exhibited reduced ERK activity (Fig. S1C).

Since the MRTFs act in partnership with SRF, we also analysed $Srf^{-/-}$ MEFs. These cells behaved similarly to $Mrtfab^{-/-}$ MEFs, exhibiting defective proliferation, increased SA-βGal and p27 expression, and upregulation of SASP markers (Fig. S1D–H). Taken together, these results show that inactivation of the MRTFs or SRF induces a non-proliferative state, some of whose molecular features are shared with classical senescence, whereas others are typical of reversible cell cycle exit and quiescence.

### MRTFs are required for proliferation of primary fibroblasts and epithelial cells
To investigate whether the proliferative defect phenotype observed in immortalised $Mrtfab^{-/-}$ MEFs was also seen in primary fibroblast cells, we isolated lung and kidney fibroblasts from $Mrtfa^{-/-}$;$Mrtfb^{fl/fl}$; $Rosa26^{Tam-Cre}$ or WT $Rosa26^{Tam-Cre}$ animals and treated them with 4OHT to induce MRTF inactivation (Fig. 2A). $Mrtfab^{-/-}$ lung and kidney fibroblasts both exhibited a clear proliferative defect compared with their WT counterparts (Fig. 2B) and also stained positive for SA-βGal (Fig. 2C). The lung fibroblasts behaved similarly to the immortalised MEF pools, with the p27 CKI and the SASP regulator TLR2 upregulated (Fig. 2D), and displayed increased levels of $Hgf$, $Mmp3$ and $Il6$ mRNAs (Fig. 2E). In contrast, kidney fibroblasts displayed upregulation of markers conventionally associated with oncogene-induced senescence – including phospho-p53 and the CKIs p21 and p16 – and upregulated the SASP components $Tlr2$, $Btg2$, $Irak1$, $Mmp9$, $Il6$, and $Il1a$ (Fig. 2D,E).

We also examined the role of MRTF–SRF signalling in proliferation of epithelial cells. Primary tracheal epithelial cells (MTECs) were isolated from $Mrtfa^{-/-}$;$Mrtfb^{fl/fl}$;$Rosa26^{Tam-Cre}$ or WT $Rosa26^{Tam-Cre}$ animals using an established protocol (Crotta et al., 2023; You et al., 2002), treated with 4OHT and plated on Transwell plastic inserts (Fig. S2A). Upon culture under non-differentiating conditions, both WT and $Mrtfab^{-/-}$ cells formed a non-ion-permeable barrier, as judged by trans-epithelial resistance, although that formed by $Mrtfab^{-/-}$ cells appeared somewhat weaker (Fig. S2B). Strikingly, however, although WT cells underwent several divisions after this point, $Mrtfab^{-/-}$ cells greatly reduced proliferation (Fig. S2C), accumulating predominantly in G2/M with increased SA-βGal activity (as assayed using CESG) and SASP marker expression (Fig. S2D–F). This was accompanied by elevated p21 expression and decreased p27 expression, and CDK1 and CKS2 levels were reduced, as seen in the fibroblasts (Fig. S2G). MRTF–SRF signalling is thus required for effective proliferation of both primary fibroblasts and epithelial cells.

### The $Mrtfab^{-/-}$ proliferation defect can be reversed by MRTF-A re-expression
Proliferative defects, accompanied by elevated SA-βGal activity and SASP marker expression, are phenotypes generally associated with

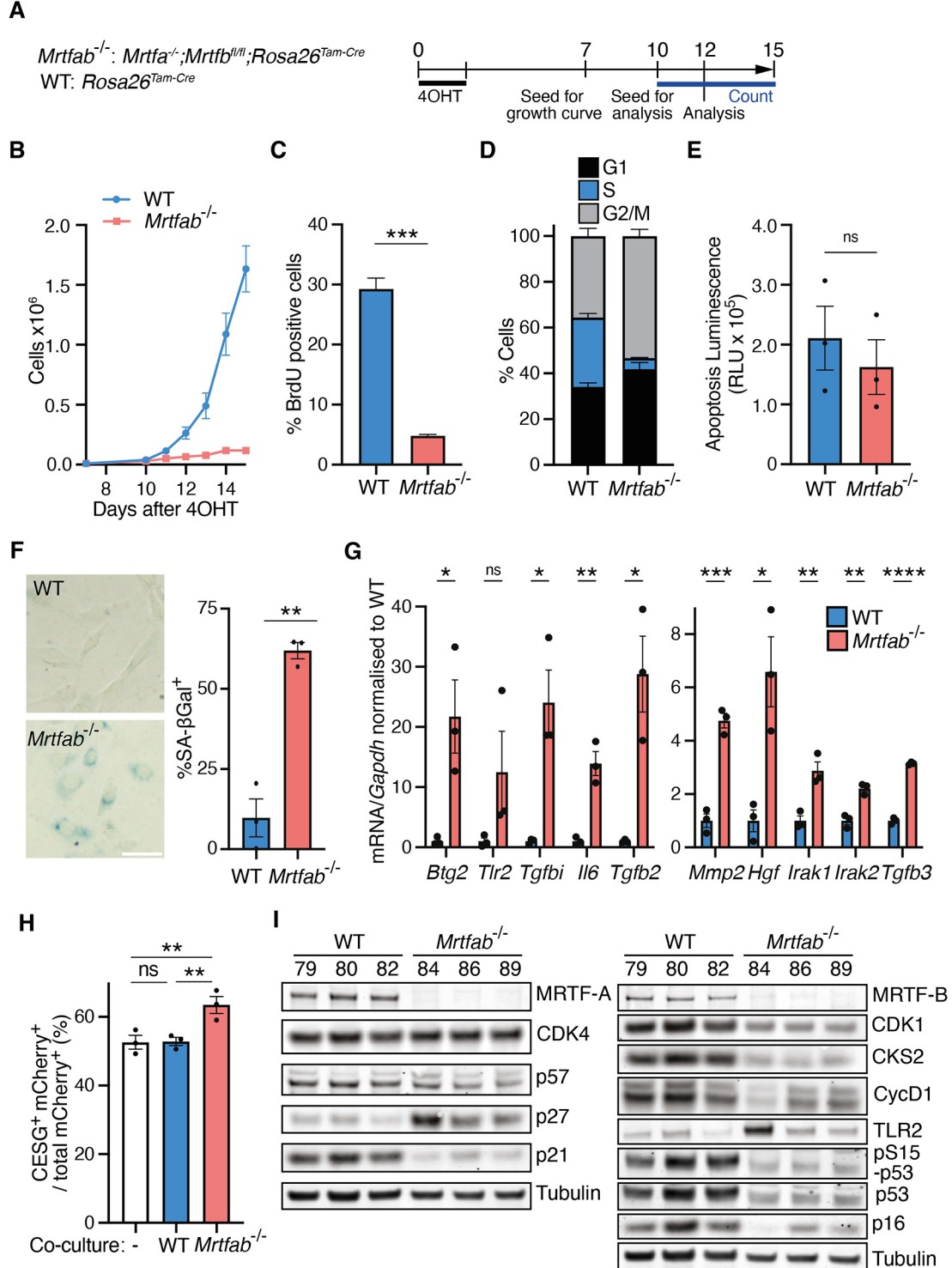

**Fig. 1.** See next page for legend.

cell senescence, which is conventionally regarded as irreversible. We therefore next tested whether the phenotypes of $Mrtfab^{-/-}$ MEFs could be reversed by MRTF-A re-expression. Clonal cell lines expressing doxycycline (Dox)-inducible HA-tagged MRTF-A together with an IRES–GFP expression marker, were derived from $Mrtfa^{-/-};Mrtfb^{fl/fl};Rosa^{Tam-Cre}$ clone 86 MEFs cells by lentiviral infection, and analysed with and without Dox and/or 4OHT treatment (hereafter $Mrtfa^{-/-};Mrtfb^{fl/fl}$ DoxMRTF-A cells;

Fig. 3A). Treatment with 4OHT, which inactivates endogenous $Mrtfb$, impaired $Mrtfa^{-/-};Mrtfb^{fl/fl}$ DoxMRTF-A proliferation, as expected, but this did not occur when MRTF-A expression was induced by Dox treatment 7 days later (Fig. 3A). MRTF-A re-expression partially suppressed the increases in TLR2 and p27 expression, and the decreases in CDK1 and CKS2 expression (Fig. 3B). It also reduced SA-βGal activity (Fig. 3C) and transcription of SASP markers (Fig. S3A).

**Fig. 1. MRTF inactivation induces a proliferation defect in SV40-immortalised MEFs.** (A) Experimental protocol. Three independent pools of SV40-immortalised MEFs were established from either $Mrtfa^{-/-}$ ($Mrtfa^{-/-}$; $Mrtfb^{fl/fl}$;$Rosa26^{Tam-Cre}$) or WT ($Rosa26^{Tam-Cre}$) animals (three embryos per genotype: $Mrtfab^{-/-}$, pools 84, 86 and 89; WT, pools 79, 80 and 82). Each pool was treated with 4OHT for 2 days, and seeded either 7 days later for growth curves or 10 days later for further analyses at day 12. (B) Cell proliferation analysed following plating of 10,000 cells on day 7 after 4OHT treatment. For each pool, cell counts at each time point are the mean of three technical replicates. Data points are the mean cell counts of the three independent cell pools±s.e.m. A representative of at least three independent experiments is shown. (C) Quantification of BrdU-positive cells in each of the three independent cell pools as measured by flow cytometry after a 2 h BrdU pulse on day 12 after 4OHT treatment. Mean±s.e.m. Significance, unpaired two-tailed $t$-test (***$P<0.001$). Data are representative of two independent experiments. (D) Cell cycle distribution of day 12 cells evaluated by PI and BrdU staining. Data are mean values from the three pools±s.e.m. Statistical significance, Fisher's LSD test: G1, $P<0.05$; S, $P<0.0001$; G2/M, $P<0.01$. Data are representative of two independent experiments. (E) Quantification of apoptosis in day 12 WT and $Mrtfab^{-/-}$ MEFs by caspase 3/7 luminescence assay. Showing data from three independent experiments, with 2–3 biological replicates in each, and 2–9 technical replicates per biological replicate. Mean ±s.e.m. Significance, unpaired two-tailed $t$-test (ns, not significant). RLU, relative luminescence units. (F) SA-βGal staining of day 12 WT and $Mrtfab^{-/-}$ MEFs. Left, representative images of each genotype (pools 80 and 86). Scale bar: 50 µm. Right, mean scores of the three pools (>50 cells scored per pool)±s.e.m. Statistical analysis, unpaired two-tailed $t$-test (**$P<0.01$). A representative of three independent experiments is shown. (G) RT-qPCR of mRNA levels of SASP factors in day 12 WT or $Mrtfab^{-/-}$ MEFs after 4OHT. Data are mean of the three independent pools (each carried out in three technical replicates)±s.e.m. Significance, unpaired two-tailed $t$-test (ns, not significant; *$P<0.05$; **$P<0.01$; ***$P<0.001$; ****$P<0.0001$). Data shown are representative of three independent experiments. (H) Primary kidney fibroblasts expressing mCherry were cultured either alone or at 100:1 with WT or $Mrtfab^{-/-}$ MEFs and cultured for 5 days before staining with CESG and analysis by flow cytometry. Data are mean values for co-culture with no cells, or with each of the three independent pools of each genotype, ±s.e.m. Statistical significance, one-way ANOVA with Fisher's LSD tests for pairwise comparisons (ns, not significant; **$P<0.01$). Data shown are representative of two independent experiments. (I) Immunoblot analysis of selected cell cycle and senescence markers in each pool of WT or $Mrtfab^{-/-}$ MEFs at day 12 after 4OHT. A representative experiment is shown, displaying data from two immunoblots. Mean marker expression (relative to WT)±s.e.m. for the three $Mrtfab^{-/-}$ pools was: CycD1, 0.84±0.29; CDK4, 1.98±0.15; TLR2, 9.52±5.55; pS15-p53 (p53 phospho-S15), 0.96±0.02; p53, 0.38±0.03; CDK1, 0.59±0.05; CKS2, 0.56±0.06; p57, 3.53±0.37; p27, 8.78±1.84; p21, 0.39±0.10; p16, 0.67 ±0.26. Spearman correlations for pairwise comparisons of marker expression were all >0.94 between WT pools and >0.87 between the $Mrtfab^{-/-}$ pools. Results are a representative from at least two independent experiments (see Fig. S1A).

We explored the effect of MRTF-A re-expression and shut-off at different times after MRTF inactivation (Fig. 3D). Re-expression of MRTF-A at 12 or even 20 days after MRTF inactivation restored p27, TLR2, CDK1 and CKS2 levels to those seen in untreated cells (Fig. 3D, samples 3 and 6). Moreover, these effects were themselves reversible: the changes induced by MRTF-A re-expression at 12 days could be reversed by Dox washout at day 17 (Fig. 3D, sample 5). MRTF-A re-expression also suppressed SA-βGal activity and SASP marker expression (Fig. S3B,C). These results show that the proliferation defect is reversible and thus distinct from classical senescence, and that MRTF-A functions redundantly with MRTF-B to promote cell proliferation.

## Serum-deprivation induces a similar proliferative defect to MRTF inactivation

The reversibility of the proliferation defect associated with MRTF inactivation, and its association with p27, a marker of reversible cell cycle arrest, led us to investigate its relationship with the proliferative arrest induced by serum deprivation. WT or $Mrtfa^{-/-}$; $Mrtfb^{fl/fl}$ cells were treated with 4OHT and then cultured in 0.3% or 10% serum prior to cell proliferation analysis (Fig. 4A). Upon culture in 0.3% serum, WT cell proliferation became comparable to that of $Mrtfab^{-/-}$ cells cultured in 10% serum, whereas $Mrtfab^{-/-}$ proliferation became virtually undetectable (Fig. 4B). After 5 days of serum deprivation, WT MEFs exhibited substantially increased expression of p27, and decreased CDK1, CKS2 and p53 phospho-S15 to levels similar to those seen in $Mrtfab^{-/-}$ MEFs maintained in 10% serum (Fig. 4C). Similar but less pronounced changes were seen after 2 days of serum deprivation (Fig. S4A). Serum deprivation induced SA-βGal activity in WT cells and significantly increased it in $Mrtfab^{-/-}$ cells (Fig. 4D). Moreover, SASP factor expression was also increased in serum-deprived WT cells, as assessed by reverse transcription quantitative PCR (RT-qPCR; Fig. S4B). Both quiescence and senescence are associated with global decreases in protein translation (reviewed by Marescal and Cheeseman, 2020; Payea et al., 2021). We found that $Mrtfab^{-/-}$ cells cultured in 10% serum displayed a diminished protein synthesis rate compared to WT MEFs, as assessed by O-propargyl-puromycin (OPP) incorporation; in both cases, culture in 0.3% serum reduced global protein synthesis rate further, to a similar basal level (Fig. 4E).

Previous studies have demonstrated that p27 inactivation does not affect the density at which primary MEFs arrest in culture, or their sensitivity to serum deprivation (Nakayama et al., 1996). We therefore tested whether inactivation of p27 could relieve the proliferation defect in $Mrtfab^{-/-}$ MEFs. We used CRISPR-Cas9 to inactivate p27 in $Mrtfa^{-/-}$;$Mrtfb^{fl/fl}$ cells, evaluating proliferation and marker expression with and without 4OHT treatment. Inactivation of p27 did not relieve the defective proliferation seen upon MRTF inactivation (Fig. 4F,G), and as seen in $Mrtfab^{-/-}$ MEFs, $p27^{-/-}Mrtfab^{-/-}$ MEFs exhibited reduced CDK1 expression, elevated SA-βGal and SASP factor expression (Fig. 4G; Fig. S4C,D). The CKI p21 is also implicated in MRTF–SRF controlled proliferation, at least in the settings described above, and p27 is at least to some extent functionally redundant with p57 (CDKN1C) in cell cycle control (Matsumoto et al., 2011; Susaki et al., 2009). Nevertheless, inactivation of all three CKIs in $Mrtfa^{-/-}Mrtfb^{fl/fl}$ cells also failed to relieve the proliferation defect resulting from MRTF inactivation (Fig. 4H). Conversely, overexpression of CDK1 or CKS2 using the doxycycline-inducible retroviral transduction strategy used for MRTF-A re-expression failed to rescue proliferation of $Mrtfab^{-/-}$ MEFs, as did expression of myoferlin, an MRTF–SRF target gene that suppresses oncogene-induced senescence in liver cancer cells (Fig. S4E; see Hermanns et al., 2017).

Taken together with the results in the preceding section, these data show that the proliferative defect in $Mrtfab^{-/-}$ MEFs is substantially similar to that induced in WT MEFs by serum deprivation, and that the altered expression of cell cycle regulators associated with it is likely to be its consequence rather than its cause.

## Cytoskeletal and cell cycle genes are downregulated in $Mrtfab^{-/-}$ MEFs

MRTF–SRF signalling plays a central role in regulation of cytoskeletal gene expression (Esnault et al., 2014; Olson and Nordheim, 2010; Schratt et al., 2002). We used RNAseq to analyse differentially expressed polyadenylated mRNAs in immortalised WT and $Mrtfa^{-/-}$; $Mrtfb^{fl/fl}$ MEFs at 0, 2, 4 and 12 days following treatment with 4OHT (referred to as T0, T2, T4 and T12 cells; Fig. S5A). Differential gene expression analysis showed that before 4OHT treatment, $Mrtfa^{-/-}$; $Mrtfb^{fl/fl}$ MEFs, which express MRTF-B alone, exhibited a very similar

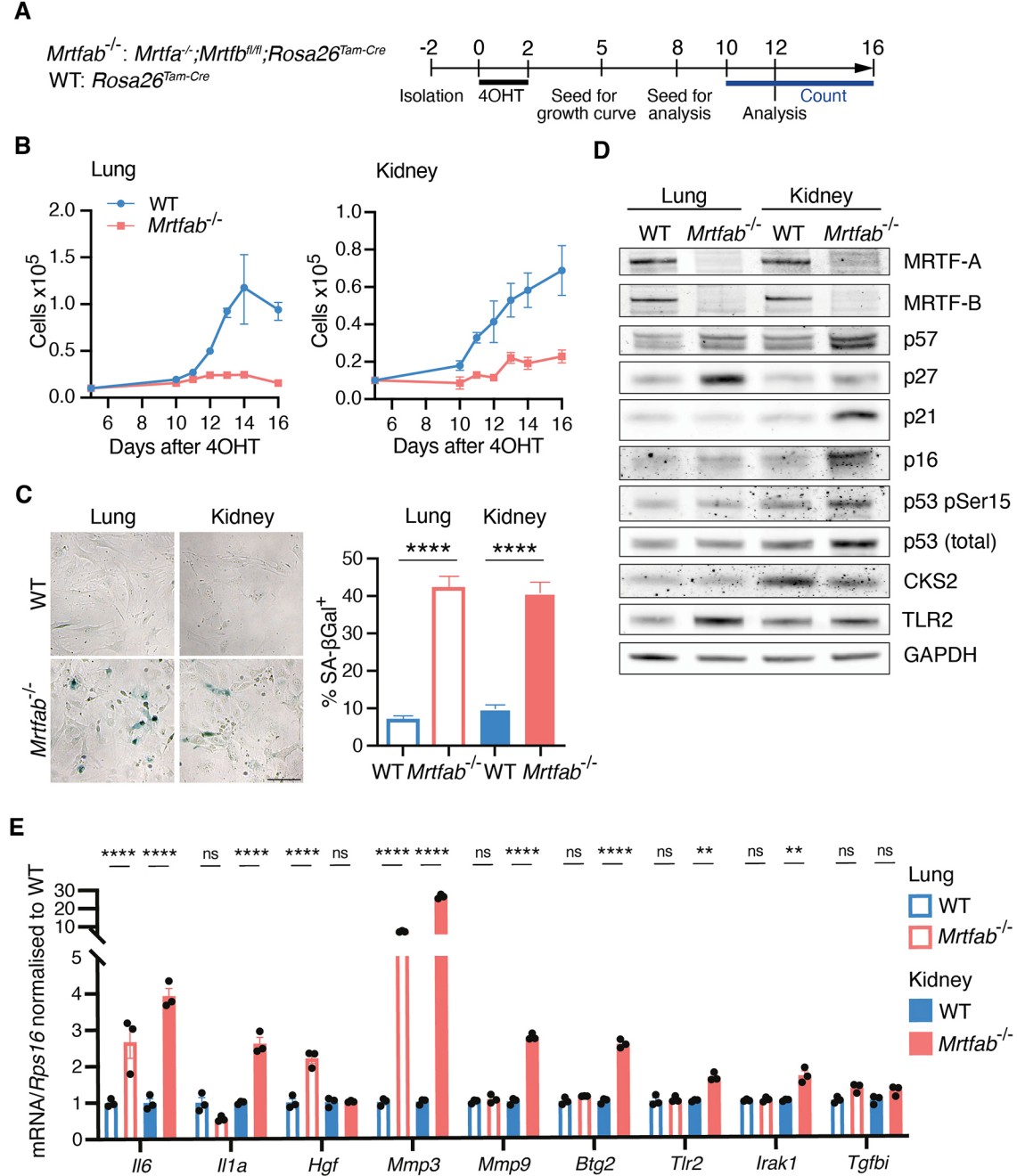

**Fig. 2. Defective proliferation of MRTF-null primary tissue-derived fibroblasts.** (A) Experimental protocol. Fibroblast pools were isolated from lung and kidney tissue of single WT or *Mrtfa⁻/⁻;Mrtfb^fl/fl* mice and treated 2 days later with 4OHT, before seeding for analysis as indicated. (B) For each cell type, 10,000 cells were seeded on day 5 after 4OHT and counted from day 10 to 16. *N*=3 mice. Data points are the mean cell counts of three independent isolates of each genotype±s.e.m. One representative example of three independent experiments is shown. (C) SA-βGal staining on day 12 after 4OHT. Left: representative images. Scale bar: 50 μm. Right, mean scores of five isolates per genotype (>100 cells scored for each isolate)±s.e.m. with one-way ANOVA with Fisher's LSD test for pairwise comparisons for statistical analysis (****$P$<0.0001). (D) Immunoblotting of lung and kidney fibroblasts on day 12 after 4OHT treatment. Representative of three independent experiments. (E) RT-qPCR of mRNA levels of SASP factors in lung and kidney fibroblasts on day 12 after 4OHT. Data are from a single experiment with three technical replicates±s.e.m., with two-way ANOVA with Fisher's LSD test for pairwise comparisons for statistical analysis (ns, not significant; **$P$<0.01; ****$P$<0.0001). Data shown are representative of three independent experiments.

gene expression profile to WT cells [Fig. 5A; Table S1; 119 genes down, 70 genes up using a threshold of log₂ fold change=1, Benjamini–Hochberg adjusted *P*-value (*P*adj)<0.05]. These genes did not exhibit significant enrichment in gene set enrichment (GSEA) analysis (Fig. S5B).

Although little change was observed at early times after 4OHT treatment, by day 12 the resulting *Mrtfab⁻/⁻* MEFs exhibited

substantial changes in gene expression (Fig. 5A; Fig. S5C; 801 genes down, 1065 genes up; see Table S1). Significantly downregulated genes included 59 out of the 683 genes previously identified as direct MRTF–SRF-binding target genes in serum-stimulated NIH3T3 cells (*P*=2.2×10⁻⁷, hypergeometric test; Esnault et al., 2014), and 131 of 328 putative MRTF target genes whose expression in MEFs was previously identified to be repressed

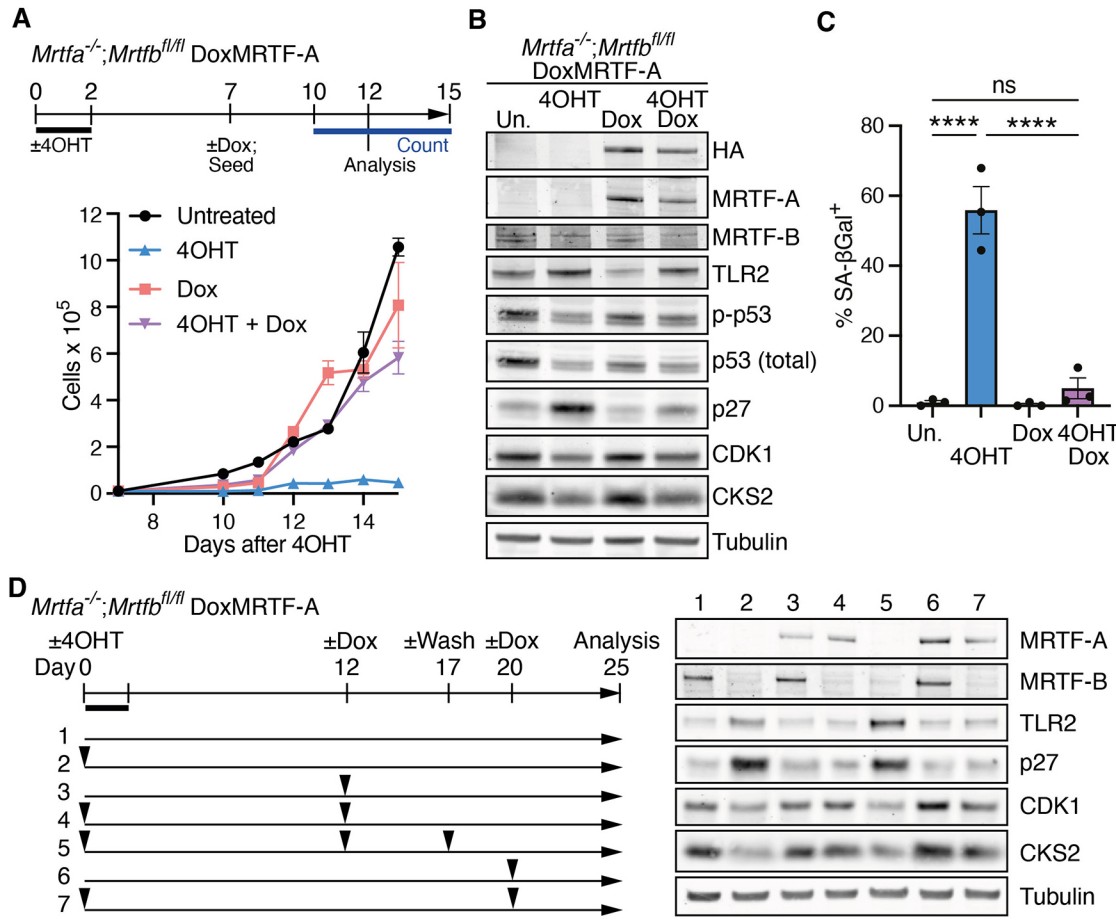

**Fig. 3. Proliferation of *Mrtfab*⁻/⁻ MEFs is reversed by MRTF-A re-expression.** (A) Top, experimental protocol. Three independent *Mrtfa*⁻/⁻*Mrtfb*^fl/fl^DoxMRTF-A clonal cell lines, derived from pool 86 *Mrtfa*⁻/⁻*Mrtfb*^fl/fl^*Rosa26*^Tam–Cre^ MEFs, were treated or not with 4OHT and/or doxycycline 7 days later. Bottom, cell proliferation following seeding of 10,000 cells of each line in triplicate at day 7. Cell counts were determined for cell line 1F2 as the mean of three technical replicates±s.e.m. Similar results were obtained with lines 1F5 and 1B2. (B) Cell cycle markers in *Mrtfa*⁻/⁻;*Mrtfb*^fl/fl^ DoxMRTF-A cell line 1F2 treated as in A at day 12 were analysed by immunoblotting (p-p53, p53 phospho-S15; Un., untreated). Similar results were obtained with lines 1F5 and 1B2. (C) SA-βGal staining of day 12 *Mrtfa*⁻/⁻*Mrtfb*^fl/fl^ DoxMRTF-A cells treated as in A at day 12. Data are mean scores of the three lines (>50 cells scored per pool)±s.e.m. Statistical analysis, one-way ANOVA with Fisher's LSD test for pairwise comparisons (ns, not significant; ****P<0.0001). (D) Extended MRTF-A activation and shut-off. Left, experimental protocol. *Mrtfa*⁻/⁻*Mrtfb*^fl/fl^ DoxMRTF-A-HA cells were treated or not with 4OHT and/or doxycycline, and/or doxycycline washout, at the indicated times. Right, immunoblotting to assess cell cycle marker expression in line 1F2 at day 25. Similar results were obtained with line 1F5.

by the TCF cofactors ($P=1.9 \times 10^{-13}$, hypergeometric test; Gualdrini et al., 2016); a substantial number of genes (32/801) were found in all three datasets (Fig. 5B). Gene ontology (GO) analysis revealed downregulation of genes involved in the structure and regulation of the actin cytoskeleton and its functions, such as adhesion and migration (Fig. 5B,C; Table S2). In addition, T12 *Mrtfab*⁻/⁻ cells downregulated genes associated with GSEA Hallmarks involved in cell proliferation, cell cycle progression, and growth (Fig. 5D, Table S2).

Many of the cytoskeletal genes identified in our analysis previously have been identified as candidate direct MRTF–SRF targets (Fig. 5B; Table S1; Esnault et al., 2014; Gualdrini et al., 2016). Notably, previous studies have implicated β and γ cytoplasmic actins, both of which are direct MRTF–SRF targets, in cell proliferation (Bunnell et al., 2011; Patrinostro et al., 2017; Tondeleir et al., 2012; see below and Discussion). In liver cancer cells, the MRTF–SRF target gene myoferlin promotes cell proliferation by suppressing oncogene-induced senescence (Hermanns et al., 2017), but myoferlin expression was not affected in our MRTF-null MEFs (Table S1).

Previous studies have suggested that MRTF–SRF signalling suppresses cell state plasticity (Hu et al., 2019; Zhang et al., 2023). Consistent with this, T12 *Mrtfab*⁻/⁻ cells exhibited upregulation of gene sets related to adipogenesis and Wnt signalling (Fig. 5C,D), and upregulation of the adipogenic markers in T12 *Mrtfab*⁻/⁻ cells was confirmed using RT-qPCR (Fig. S5D). Adipocytes could not be detected in T12 *Mrtfab*⁻/⁻ cells by Oil Red O staining, however, although both WT and T12 *Mrtfab*⁻/⁻ cells could differentiate into fat cells under adipogenic culture conditions (Fig. S5E). Thus, T12 *Mrtfab*⁻/⁻ MEFs appear to have relaxed cell state plasticity (see Discussion).

### Altered morphology and cytoskeletal dysfunction in *Mrtfab*⁻/⁻ MEFs
We next investigated the relationship between defective cytoskeletal gene expression in *Mrtfab*⁻/⁻ MEFs and proliferation. Consistent with studies in other systems (Alberti et al., 2005; Medjkane et al., 2009; Schratt et al., 2002), *Mrtfab*⁻/⁻ MEFs exhibited a rounded morphology and lacked well-defined stress fibres and focal adhesions, which were replaced by cortical paxillin-rich puncta

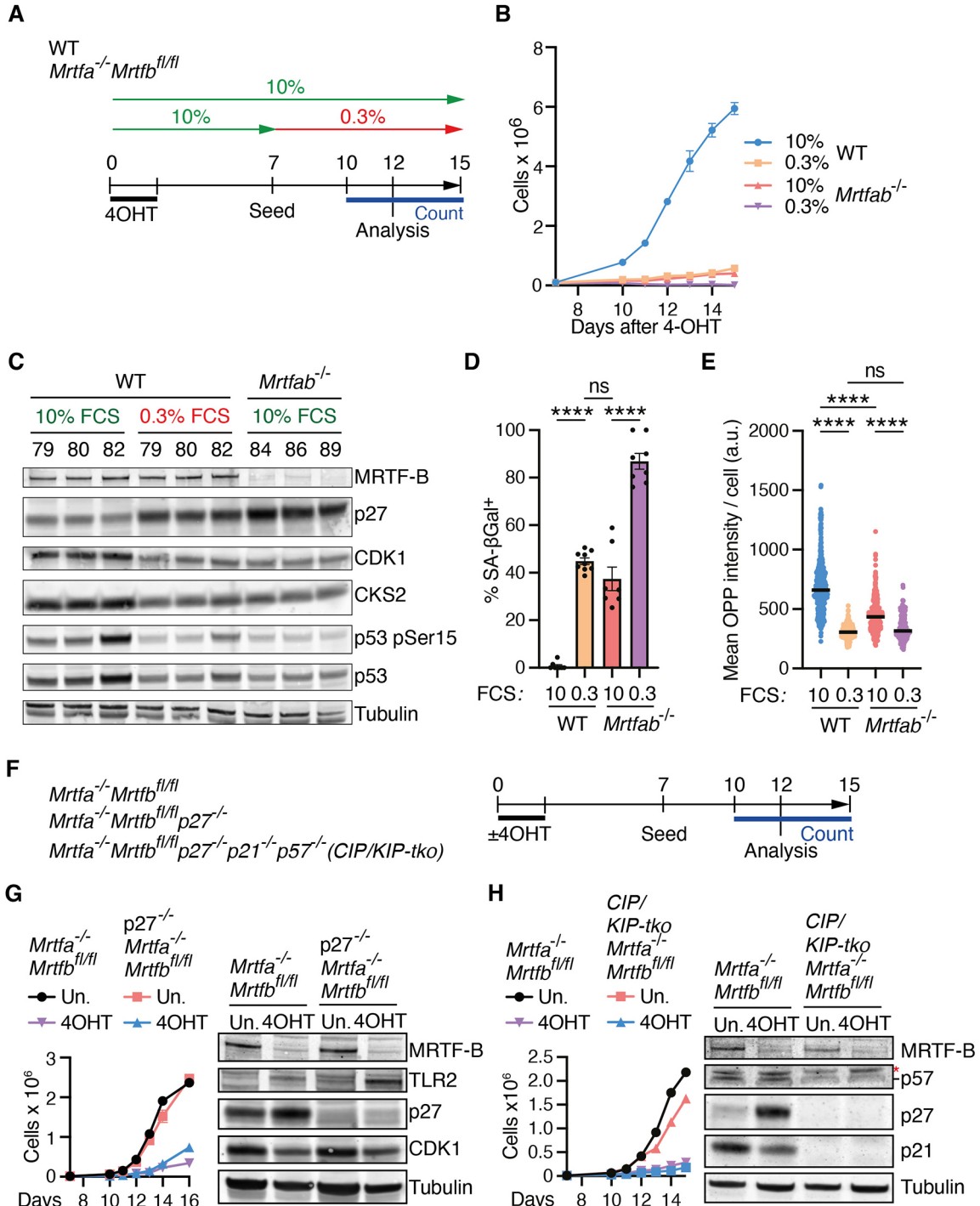

**Fig. 4. The MRTF-null phenotype exhibits features of quiescence.** (A) Experimental protocol. The three pools of WT and *Mrtfab*[−/−] MEFs were treated with 4OHT and cultured in 10% FCS, with medium replaced or changed to 0.3% FCS on day 7. (B) Proliferation of cells treated as in A. For each experimental group, 10,000 cells were seeded on day 7. Data are mean±s.e.m. of the three pools for each genotype. One of two independent experiments is shown. (C) The three pools of WT and *Mrtfab*[−/−] MEFs were treated as in A, and cell cycle markers were analysed by immunoblot at day 12. One of two independent experiments is shown. (D) WT (pool 80) and *Mrtfab*[−/−] (pool 86) MEFs were treated as in A, and SA-βGal staining was analysed at day 12. Data are mean±s.e.m., *n*=7–9 replicates per line, with one-way ANOVA with Fisher's LSD test for statistical analysis (ns, not significant; ****$P<0.0001$). (E) WT (pool 80) and *Mrtfab*[−/−] (pool 86) MEFs were treated as in A, and protein biosynthesis was measured by OPP incorporation at day 12 (a.u., arbitrary units). Data are values for individual cells (>170 cells per condition/genotype), with mean indicated. One-way ANOVA with Tukey's post-hoc test was used for statistical analysis (ns, not significant; ****$P<0.0001$). One representative example of three independent experiments is shown. (F) p27-null and CIP/KIP triple knockout (tko) *Mrtfa*[−/−]*Mrtfb*[fl/fl] MEFs were generated from *Mrtfa*[−/−];*Mrtfb*[fl/fl] pool 86 MEFs by CRISPR-Cas9 mutagenesis. Following treatment or not with 4OHT, 10,000 cells were seeded on day 7, with immunoblot analysis at day 12. (G) Growth curve and cell cycle markers for p27[−/−]*Mrtfa*[−/−]*Mrtfb*[fl/fl] MEFs treated as in F (Un., untreated). Data for clone 3A2 are shown. Similar results were obtained with clones 1F3 and 3B10. (H) Growth curve and cell cycle markers for p27[−/−]p21[−/−]p57[−/−]*Mrtfa*[−/−]*Mrtfb*[fl/fl] MEFs treated as in F (Un., untreated). Data for clone 3.1D10 are shown. Similar results were obtained with clones 3.1C11 and 3.1G9. Asterisk, unspecific band.

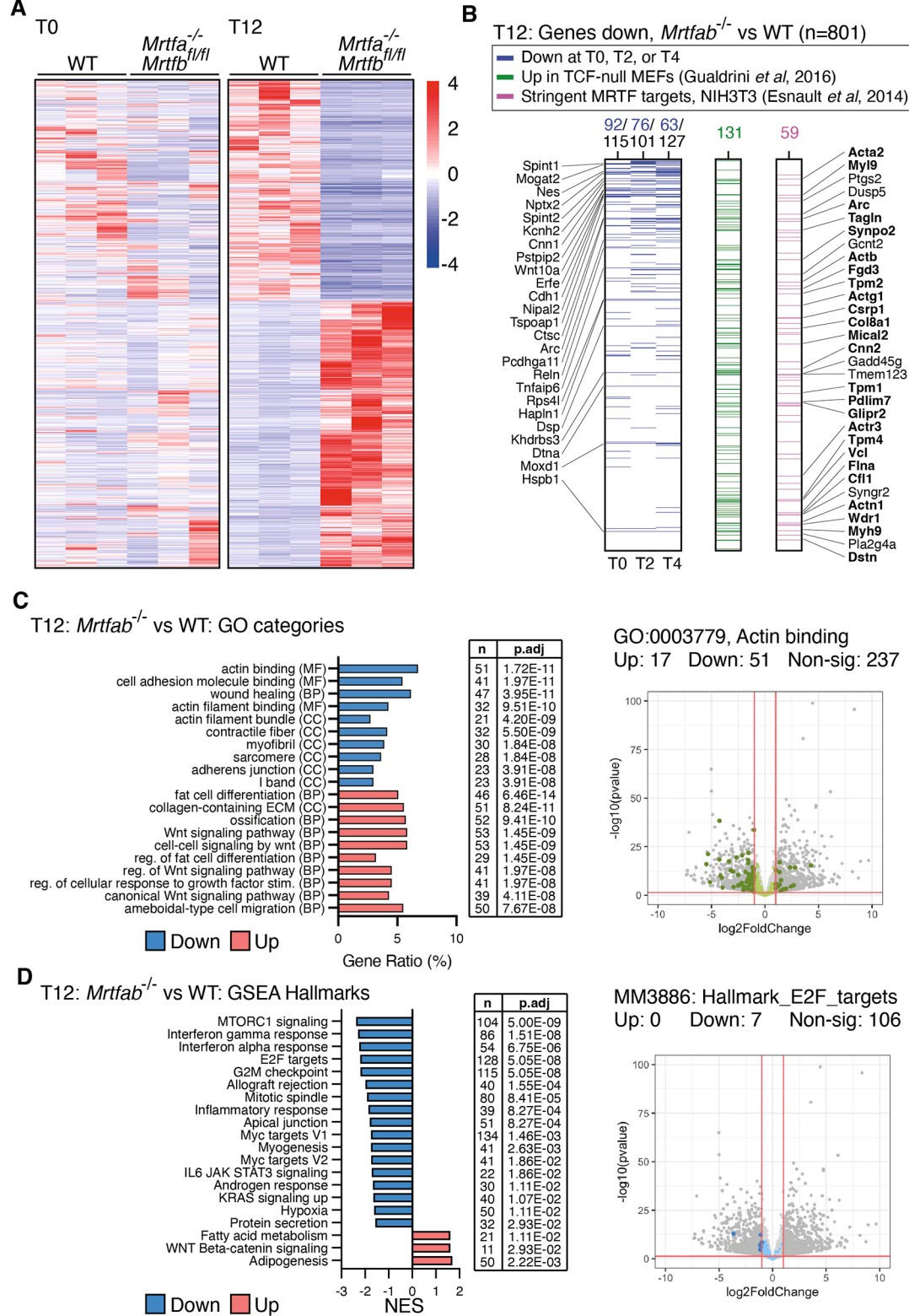

**Fig. 5.** See next page for legend.

(Fig. 6A). F-actin levels, as assessed by phalloidin staining, were significantly reduced (Fig. 6B), and total actin levels were substantially decreased, as was the F/G-actin ratio (Fig. 6C). MRTF-A re-expression in *Mrtfab*$^{-/-}$ MEFs was sufficient to restore normal cell morphology and F-actin distribution (Fig. S6A). Phasefocus

Livecyte microscopy revealed that *Mrtfab*$^{-/-}$ MEFs occupy a reduced cell area, with increased sphericity and cell thickness, but their volume was unchanged (Fig. 6D). Upon plating, *Mrtfab*$^{-/-}$ MEFs spread more slowly than WT cells, and more isotropically (Fig. S6B,C). They were defective in adhesion to the integrin ligands fibronectin and

**Fig. 5. Downregulation of cytoskeletal and cell cycle genes in *Mrtfab*−/− MEFs.** (A) WT MEF pools 79, 80 and 82; and *Mrtfa*−/−*Mrtfb*fl/fl MEF pools 84, 86 and 89 were treated or not with 4OHT and cultured for 0, 2, 4 or 12 days before RNAseq analysis (T0, T2, T4, T12 samples; Table S1; see Fig. S5A). Heatmap shows genes differentially expressed at T12 (>2-fold change and *P*adj<0.05 in the comparison between WT and *Mrtfab*−/− MEFs at T12). z-scores of genes for T0 and T12 cells are colour-coded as indicated and ordered using unsupervised clustering. (B) Relationships between the 801 genes with gene symbols that map to an ensembl ID and are significantly downregulated (>2-fold change, *P*adj<0.05) in *Mrtfab*−/− MEFs on day 12. The genes are ranked by fold-change upon MRTF inactivation. Left, genes downregulated in *Mrtfab*−/− MEFs at other timepoints (blue), with those common to all timepoints listed. Centre, identity with genes upregulated in MEFs upon inactivation of the TCFs (green) (Gualdrini et al., 2016). Right, identity with the direct MRTF–SRF target genes stringently defined in NIH3T3 cells (magenta) (Esnault et al., 2014); the list shows genes also upregulated upon TCF inactivation, with cytoskeletal components and regulators in bold. (C) GO category analysis of differentially regulated genes in *Mrtfab*−/− MEFs on day 12. Bar plots, top 10 downregulated (blue) and upregulated (red) GO terms from any category (BP, biological process; MF, molecular function; CC, cellular component) ranked according to *P*adj. Table, number of genes changing, and the Benjamini–Hochberg adjusted *P*-value for each gene set. Volcano plot for the specimen GO term GO:0003779 Actin binding is shown on the right, with fold change and *P*adj thresholds indicated. Dark green, significant genes of the GO term. Dark grey, other significant genes. Light green, non-significant genes of the GO term. Light grey, other non-significant genes. (D) GSEA Hallmark analysis of differentially regulated genes in *Mrtfab*−/− MEFs on day 12. Bar plots, top 20 GSEA hallmark gene sets, showing the normalised enrichment score (blue, downregulated; red, upregulated). Table, number of genes changing and the Benjamini–Hochberg adjusted *P*-value for each gene set. Volcano plot for the specimen GSEA Hallmark MM3886: HALLMARK_E2F_TARGETS is shown at right with fold change and *P*adj thresholds indicated. Dark blue, significant genes of the GO term. Dark grey, other significant genes. Light blue, non-significant genes of the GO term. Light grey, other non-significant genes. Non-sig, not significant.

vitronectin, but not to poly-L-lysine (PLL), which allows integrin-independent cell adhesion, consistent with their lack of focal adhesions (Fig. S6D). *Mrtfab*−/− MEFs also exhibited defective motility, with significantly reduced instantaneous velocity compared to that of WT cells and little net displacement over the 96 h monitoring period (Fig. 6E; Movies 1,2). MRTF-A re-expression in *Mrtfab*−/−DoxMRTF-A cells increased cell velocity and net displacement, and decreased cell sphericity (Fig. S6E,F; Movies 3–8). Taken together, these data show that MRTF-A and MRTF-B function redundantly in MEFs to control actin cytoskeletal dynamics, and adhesive and motile behaviour.

### *Mrtfab*−/− MEF proliferation remains responsive to mechanical cues

Previous studies have shown that proliferation of adherent cells is dependent on both adhesion and matrix compliance, and cell tension and spreading (Chen et al., 1997; Huang et al., 1998; Klein et al., 2009; Mammoto et al., 2004, reviewed by Mammoto and Ingber, 2009). The multiple cytoskeletal defects seen in *Mrtfab*−/− MEFs raises the possibility that their defective proliferation might result from inability to respond to such mechanical environmental cues. First, we examined the effect of matrix compliance. WT and *Mrtfab*−/− MEFs were plated on substrates of increasing rigidity, and proliferation was measured by 5-ethynyl-2′-deoxyuridine (EdU) incorporation. Proliferation of WT cells was substantially reduced on soft substrates, as expected, to a rate comparable to that of *Mrtfab*−/− cells on hard substrates; however, *Mrtfab*−/− cell proliferation was further reduced on soft substrates (Fig. 7A). Next, we examined the effects of cell spreading. Proliferation of WT cells

was reduced upon plating on patterns that limit cell–substrate contact area (Fig. 7B) but was unaffected when cells were grown on pillars, which restrict the total adhesion area but allow cells to spread normally (Fig. 7C). Proliferation of *Mrtfab*−/− cells was similar to that seen for WT cells under conditions of limiting adhesion area; however, they remained sensitive to confinement, indicating that other pathways might also contribute (see Discussion).

### Defective *Mrtfab*−/− MEF proliferation reflects impaired actomyosin contractility

Given the role of MRTF–SRF signalling in regulation of cytoskeletal dynamics, we next investigated how direct interference with cytoskeletal dynamics affects cell cycle progression. Previous studies have shown that Rho signalling both potentiates focal adhesion assembly and relieves p27-dependent cell cycle arrest (Hotchin and Hall, 1995; Mammoto et al., 2004). However, following 4OHT treatment, the proliferation of *Mrtfa*−/−;*Mrtfb*fl/fl cells was not restored upon expression of constitutively active RhoA (RhoAG14V; Fig. S7A,B). We therefore investigated whether interference with cytoskeletal dynamics in WT MEFs could affect cell cycle progression. We focussed on cell contractility, which is required for progression through the G1/S checkpoint in endothelial cells (Huang et al., 1998). ROCKs are important regulators of cell contractility, controlling both F-actin assembly and myosin activity (reviewed by Rath and Olson, 2012), and ROCK inhibition has previously been reported to lead to both cytokinesis defects and cellular senescence (Kumper et al., 2016).

Treatment of WT MEFs with the pan-ROCK inhibitor H1152 reduced actin stress fibres and focal adhesions, as did serum deprivation, but did not induce the cell rounding seen in *Mrtfab*−/− MEFs (Fig. S7C). Nevertheless, H1152 reduced proliferation of immortalised WT MEFs to a level comparable to that of *Mrtfab*−/− MEFs (Fig. 7D–F). ROCK-inhibited WT MEFs exhibited similar changes in p27, CDK1 and CKS2 expression to *Mrtfab*−/− MEFs and serum-deprived WT MEFs (Fig. 7G), as well as increased SA-βGal activity and elevated transcription of SASP markers (Fig. S7E,F). In contrast to *Mrtfab*−/− MEFs, however, H1152 treatment of WT MEFs induced significant numbers of multinucleated cells (Fig. S7C,D), as expected given the role of the ROCKs in cytokinesis (Green et al., 2012). Treatment with the myosin II inhibitor blebbistatin also inhibited proliferation of WT MEFs (Fig. 7D,H). Blebbistatin-treated WT MEFs exhibited similar cytoskeletal changes to ROCK-inhibited cells (Fig. S7C), and blebbistatin treatment induced similar effects on cell cycle marker expression to H1152 treatment or serum deprivation of WT cells, or MRTF inactivation (Fig. 7G). Finally, we directly examined contractility in *Mrtfab*−/− MEFs using traction force microscopy. *Mrtfab*−/− MEFs exhibited reduced contractility, exerting 70% less force per unit area than WT MEFs (Fig. 7I). Taken together, these results suggest that the proliferation defect observed in *Mrtfab*−/− MEFs is likely due to intrinsic defects in cytoskeletal dynamics, and that this at least in part reflects reduced contractility.

### DISCUSSION

In this study we have shown that the MRTF–SRF axis plays a significant role in cell proliferation in fibroblast and epithelial cells. Inactivation of SRF, or its MRTF cofactors, induces a quiescence-like state under conditions that permit efficient cell cycle progression in WT cells. This state features decreased levels of cyclin D, CDK1 and CKS2, and increased amounts of the CKI p27, previously associated with reversible cell cycle arrest. While fully reversible, this state exhibits features associated with classical senescence, including elevated neutral β-galactosidase activity, and expression of numerous

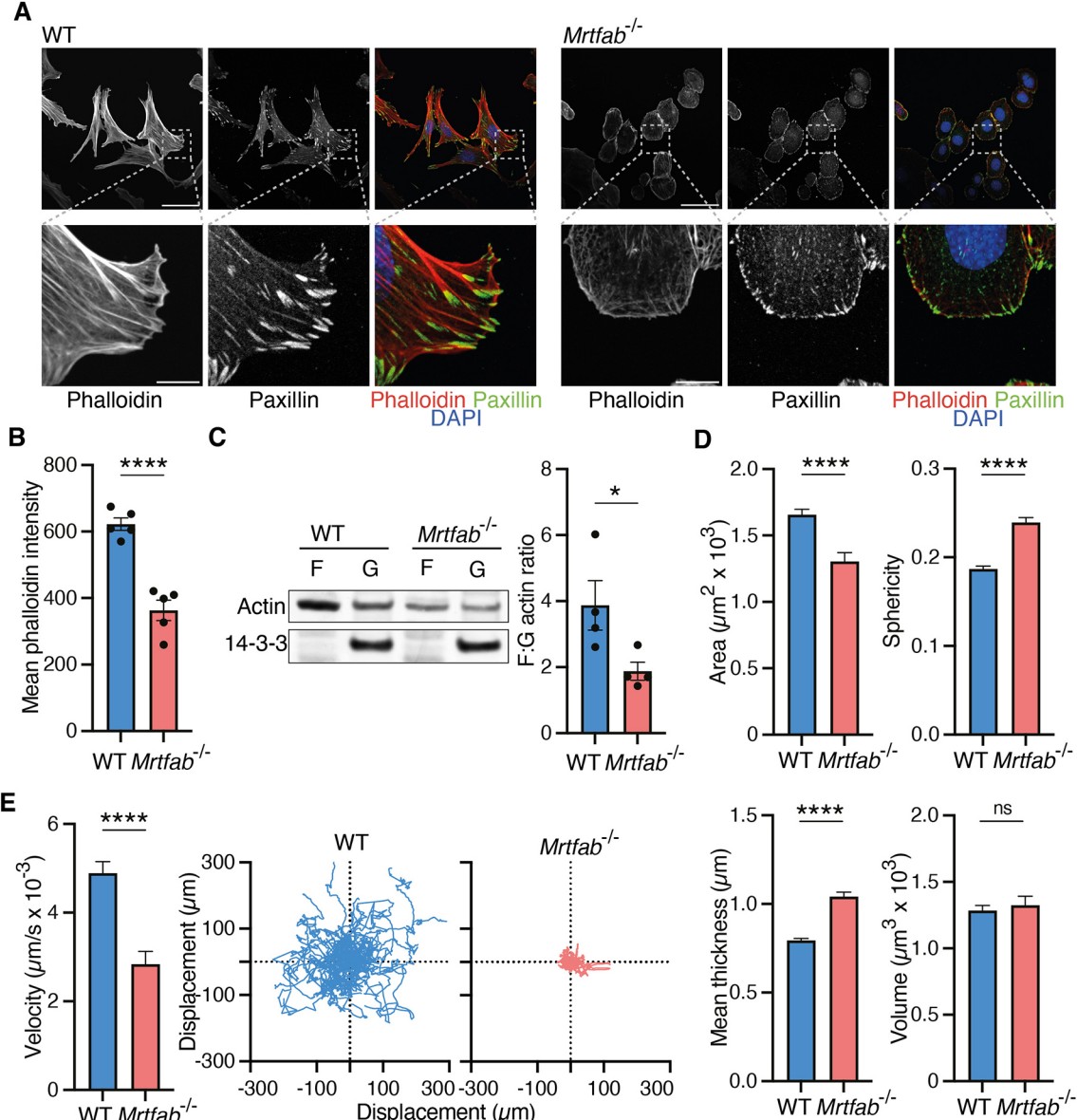

**Fig. 6. Cytoskeletal defects in *Mrtfab*⁻/⁻ MEFs.** WT or *Mrtfab*⁻/⁻ MEFs (pools 80 and 86, respectively) were analysed at day 12 after 4OHT. Similar results were obtained with the other MEF pools. (A) Immunofluorescence staining of paxillin (focal adhesions), and staining with phalloidin (labelling F-actin) and DAPI (labelling DNA). Scale bars: 50 μm (top) and 10 μm (zoomed in images, bottom). Representative of five independent experiments. (B) Mean phalloidin staining intensity per cell of cells cultured and stained as in A. Data are mean±s.e.m. of five independent experiments. Statistical analysis, two-tailed unpaired *t*-test (****$P<0.0001$). (C) Left, F-actin pelleting assay. 14-3-3, soluble protein control. Representative of four independent experiments. Right, F/G-actin ratio; data are mean±s.e.m., $n=4$; statistical analysis, two-tailed unpaired *t*-test (*$P<0.05$). (D) Morphological analysis of cell area, sphericity, thickness and volume by Phasefocus Livecyte microscopy. Measurements are from the first frame of the imaging sequence, with >100 cells per sample. Data are mean from three technical replicates±s.e.m. Statistical analysis, two-tailed unpaired *t*-test (ns, not significant; ****$P<0.0001$). (E) Analysis of cell motility of cells in D by Phasefocus Livecyte microscopy. Cells were imaged over 96 h at 2 images/h. For each genotype, 50 trajectories were monitored until cell division, exit of frame of view or end of video. Left, instantaneous velocity, plotting measurements from frame *n* to *n*+1 throughout the whole imaging sequence, with >100 cells per sample. Data are compiled from three technical replicates. Mean±s.e.m. Statistical analysis, two-tailed unpaired *t*-test (****$P<0.0001$). Right, displacement plots of cells in a representative experiment.

SASP markers. Similar phenotypes are also seen in WT cells upon serum starvation, or upon inhibition of the ROCKs or of actomyosin contractility. MRTF–SRF inactivation results in multiple cytoskeletal defects, including substantially reduced contractility. MRTF-A and MRTF-B function redundantly to promote cytoskeletal integrity and cell proliferation. Our findings suggest that MRTF–SRF-dependent cytoskeletal dynamics, particularly contractility, play a pivotal role in generating the pro-proliferative signal provided by cell–substrate adhesion in adherent cells.

Several aspects of the *Mrtfab*⁻/⁻ phenotype are seen in senescent cells, but we do not consider MRTF-null cells to exhibit classical senescence. The stringent definition of senescence has been hampered by the lack of well-defined and specific markers (Munoz-Espin and Serrano, 2014). MRTF-null MEFs exhibit a SASP-like secretory phenotype and can inhibit proliferation of co-cultured WT cells. However, SASP-like phenotypes have also been reported in quiescent cells (Anwar et al., 2018), and recent studies suggest that the SASP and SA-βGal reflect duration of cell cycle withdrawal rather than a

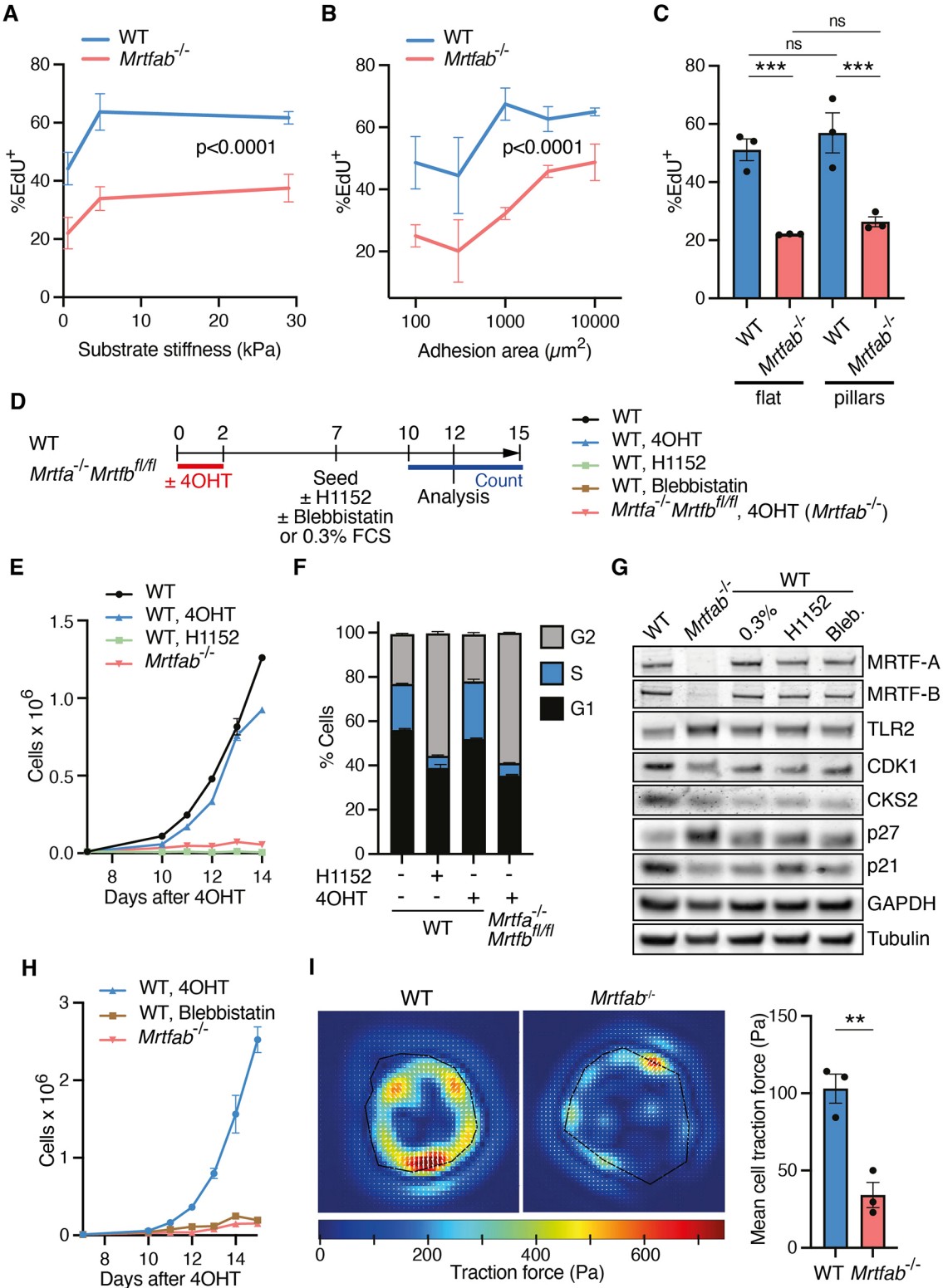

**Fig. 7.** See next page for legend.

distinct non-proliferative state (Ashraf et al., 2023). Irreversibility has conventionally been considered a hallmark of senescence (Campisi and d'Adda di Fagagna, 2007), but we found that all MRTF-null MEF phenotypes can be fully reversed by MRTF-A re-expression. Finally, although we found that $Mrtfab^{-/-}$ MEFs apparently exhibit increased cell plasticity, a phenotype associated with therapy-induced cancer

cell senescence (Milanovic et al., 2018), MRTF–SRF activity has also been previously reported to suppress plasticity in non-senescent cells (Hu et al., 2019).

Previous studies of the MRTFs and SRF have led to the view that the pathway affects cell motile and adhesive properties rather than cell proliferation. SRF-null embryos proliferate normally up to E6

**Fig. 7. Inhibition of contractility in WT cells induces a phenotype similar to MRTF inactivation.** (A) Cells from the three WT and *Mrtfab*−/− MEF pools were plated on polyacrylamide substrates of different stiffnesses at day 10 following 4OHT treatment, cultured for 2 days and cell cycle status assessed by EdU incorporation during a 6 h period on day 12. Data are mean of >100 cells per sample±s.e.m. Statistical analysis by two-way ANOVA reports significance of the main effect (WT versus *Mrtfab*−/−). One of two independent experiments is shown. (B) Cells from each WT and *Mrtfab*−/− MEF pool were plated on islands of different areas at day 10 following 4OHT treatment. Area per cell is ~1500 μm² (Fig. 6D). Cell cycle status at day 12 was analysed and presented as in A. One of two independent experiments is shown. (C) Cells from each WT and *Mrtfab*−/− MEF pool were plated at day 10 following 4OHT treatment either on flat surfaces or on etched patterned substrates (1 μm in diameter and 3 μm between features) in which only ~5% of a given area was available for adhesion (pillars). EdU incorporation at day 12 was analysed and presented as in A. Statistical analysis used one-way ANOVA with multiple comparisons using Fisher's LSD test (ns, not significant; ***P<0.001). (D) Schematic of protocol for panels E–H. *Mrtfab*−/− (*Mrtfa*−/−*Mrtfb*fl/fl*Rosa26*Tam−Cre) pool 86 or WT (*Rosa26*Tam−Cre) pool 80 cells were treated or not with 4OHT, H1152 and blebbistatin as indicated before analysis. (E) Growth curves of pool 80 and pool 86 cells treated as shown in D. For each experimental group, 10,000 cells were seeded on day 7 and counted from day 10 to 15. Data are mean of three technical replicates±s.e.m., and results are representative of three independent experiments. (F) Quantification of BrdU-positive cells as measured by flow cytometry after a 2 h BrdU pulse on day 12 after 4OHT. Data are mean of the three independent pools±s.e.m. Significance, two-way ANOVA interaction at *P*<0.0001. (G) Immunoblot analysis of cell cycle markers in day 12 WT (pool 80) or *Mrtfab*−/− (pool 86) MEFs treated as indicated. Data are from one representative example of three independent experiments. (H) Growth curves of WT (pool 80) cells treated with blebbistatin compared with *Mrtfab*−/− cells (pool 86) as in D. For each experimental group, 10,000 cells were seeded on day 7 and counted from day 10 to 15. Data are mean of three technical replicates±s.e.m. from three independent experiments. (I) The three pools of WT or *Mrtfab*−/− MEFs were plated on 30 kPa gels and analysed by traction force microscopy on day 12. Traction force was determined for 15–20 cells for each pool. Images of deduced traction force for representative cells from pool 80 and pool 86. Data are mean traction force for each pool±s.e.m., *n*=3 independent experiments. Statistical analysis, two-tailed unpaired *t*-test (**P<0.01).

but fail to gastrulate (Arsenian et al., 1998; Schratt et al., 2001), and SRF-null neuroblasts form in normal numbers but are unable to migrate to the olfactory bulb (Alberti et al., 2005). Inactivation of SRF or the MRTFs does not appreciably impair the early proliferative stages of T cell differentiation or activation (Fleige et al., 2007; Maurice et al., 2024; Mylona et al., 2011). However, in both human and mouse, smooth muscle cells that lack SRF exhibit an increased propensity to enter senescence (Angstenberger et al., 2007; Hengst et al., 1994; Werth et al., 2010), while MRTF–SRF signalling suppresses oncogene-induced senescence in cancer cells lacking the tumour suppressor DLC1, a RhoGAP (Hampl et al., 2013; Hermanns et al., 2017). Although further work is needed to elucidate the basis for these context-dependent differences, our data suggest that MRTF–SRF-dependent proliferation may be more common than previously thought, particularly when proliferation is dependent on contractility.

Elevated p27 levels are characteristic of reversible cell cycle inhibition or quiescence induced in various cell contexts by non-genotoxic cues including serum deprivation (Coats et al., 1996), TGF-β treatment or contact inhibition (Polyak et al., 1994), restriction of cell spreading (Chen et al., 1997; Huang et al., 1998), low matrix stiffness (Klein et al., 2009), or lovastatin treatment (Hengst et al., 1994; Hengst and Reed, 1996). Many of these directly affect activation of the MRTF–SRF signalling pathway or impinge on cytoskeletal dynamics. At least in some settings, arrest-induced p27 accumulation, which is Rho dependent, reflects decreased Skp2

levels (Huang et al., 1998; Mammoto et al., 2004). Decreased Skp2 levels are also seen in SRF-null vascular smooth muscle cells (Werth et al., 2010). Both Skp2 protein and transcript levels were slightly decreased in our MRTF-null MEFs, and we note that in breast cancer cells, Skp2 expression also responds to mechanical cues via the YAP pathway (Jang et al., 2017).

The CKIs p27 and p21 inhibit CDKs through a common mechanism (Polyak et al., 1994), and p27 and p57 share functions in proliferation control (Matsumoto et al., 2011; Susaki et al., 2009). We found that in contrast to immortalised *Mrtfab*−/− MEFs, *Mrtfab*−/− primary kidney fibroblasts and tracheal epithelial cells exhibited elevated p21 levels, with correspondingly reduced p27 levels. While this suggests that p27 or other CIP/KIP family proteins might function redundantly to mediate the proliferation defect, we found that inactivation of neither p27 nor all three CIP/KIP proteins was sufficient to restore proliferation to *Mrtfab*−/− MEFs. This finding extends previous studies showing that MEFs lacking individual CIP/KIP family members retain growth control by contact inhibition and serum deprivation (Deng et al., 1995; Nakayama et al., 1996; Takahashi et al., 2000). We therefore suggest that CKI elevation upon MRTF inactivation is a consequence rather than a cause of proliferation arrest in MEFs.

We found that culture of WT MEFs on limited areas of adhesion also reduced cell proliferation to a comparable extent to unconfined *Mrtfab*−/− cells. Such confinement also increases cellular G-actin levels, which would be expected to decrease MRTF activity (R. Fedoryshchak et al., unpublished). In contrast, proliferation was not affected when WT cells were allowed to spread under conditions of limited adhesion area, in agreement with previous findings (Chen et al., 1997), and in this case G-actin levels were unaffected. We therefore speculate that the primary driver for confinement-mediated impairment of cell proliferation might be altered actin dynamics. We note, however, that the greatly reduced proliferation of *Mrtfab*−/− MEFs can also be further inhibited by confinement, suggesting other mechano-responsive pathways promote proliferation. One such pathway might be the mechano-responsive YAP–TAZ system (Dupont et al., 2011).

The quiescence-like state associated with MRTF inactivation can also be induced in WT MEFs by treatment with the ROCK1 and ROCK2 inhibitor H1152 or the myosin II inhibitor blebbistatin. Both ROCKs and myosin are critical for cell contractility, and MRTF-null MEFs exhibit substantially impaired contractility. Moreover, classical studies have shown that F-actin integrity and cell contractility are required for progression through the G1 restriction point (Huang et al., 1998; Iwig et al., 1995; Kumper et al., 2016; Reshetnikova et al., 2000). How might this affect cell progression? Two non-mutually exclusive mechanisms can be envisaged. First, MRTF inactivation might directly limit contractility-dependent changes in cell morphology required for cell cycle progression. These could be cell intrinsic or imposed by the environment. For example, during G2/M, ROCK- and myosin-dependent contractility promotes mitotic cell rounding by multiple mechanisms (reviewed by Ramkumar and Baum, 2016), including focal adhesion disassembly and the assembly of rigid cortical actomyosin (Jones et al., 2018; Matthews et al., 2012; Ramanathan et al., 2015), and the actomyosin contractile ring at cytokinesis (Green et al., 2012). Moreover, in pseudo-stratified epithelia such as the tracheal epithelial cells studied here, actomyosin contractility is essential for the nuclear movements required for mitosis (reviewed by Meyer et al., 2011). External mechanical constraints arising from tissue stiffness, fibrosis or tumour microenvironment might also render cell cycle progression dependent on MRTF–SRF-dependent cytoskeletal dynamics. A

second possibility is that MRTF inactivation limits the generation of pro-proliferative signals from the cytoskeleton. Mechanical cues, including cell adhesion and spreading (Chen et al., 1997; Huang et al., 1998; Mammoto et al., 2004), and matrix compliance (Klein et al., 2009; Mih et al., 2011), promote cell cycle progression in anchorage-dependent cells. Focal adhesions, whose assembly is MRTF–SRF dependent in our cells, would be prime candidates to generate such signals (for review see Burridge and Chrzanowska-Wodnicka, 1996; Kamranvar et al., 2022).

MRTF–SRF target genes include both components and regulators of the actin cytoskeleton, including the actins themselves (Schratt et al., 2002; reviewed by Olson and Nordheim, 2010). We propose that the contractility deficits underpinning defective proliferation of MRTF-null cells reflect reduced expression of multiple cytoskeletal MRTF–SRF target genes, although it remains possible that MRTF–SRF also directly controls expression of cell cycle regulators. G-actin plays a critical role in regulation of MRTF activity, and in at least some contexts overexpression of cytoplasmic actin can rescue MRTF-dependent phenotypes (Maurice et al., 2024; Salvany et al., 2014). Intriguingly, in MEFs the inactivation of cytoplasmic actin genes results in defective proliferation (Bunnell et al., 2011; Patrinostro et al., 2017; Tondeleir et al., 2012). The proliferative and cytoskeletal phenotypes of MRTF-null cells might thus at root reflect deficient actin expression, and we are currently testing this idea.

We have shown that MRTF–SRF activity is required for proliferation of anchorage-dependent fibroblasts and epithelial cells, most likely because MRTF–SRF-dependent cytoskeletal dynamics and contractility allow adherent cells to execute the morphological changes required for cell division. However, many untransformed cell types also proliferate independently of adhesion, and anchorage-independent proliferation is also a hallmark of oncogenic transformation. Indeed, both published knockdown data and our own unpublished knockout results indicate that MRTF–SRF activity is not required for proliferation of all transformed cells (Hampl et al., 2013; Medjkane et al., 2009). In principle, oncogenic signals could simply substitute for pro-proliferative signals generated in non-transformed cells by substrate adhesion. According to this view, proliferation of transformed cells might occur independently of MRTF–SRF activity, providing that constraints on cytoskeletal dynamics are not limiting, which might well be the case for non-adherent cells. Our future work will focus on the relation between transformation, MRTF–SRF activity and the cytoskeleton.

## MATERIALS AND METHODS
### Mouse embryonic fibroblast culture
Animals used were held under UK Home Office project licence P0389970. Primary fibroblasts of wild type, $Srf^{fl/fl}$, and $Mrtfa^{-/-}Mrtfb^{fl/fl}$ were obtained from lung or kidney tissue from appropriate crosses of 'Wildtype' [R26CreERT2 ($tm9(cre/ESR1)Arte$)] (Seibler et al., 2003), conditional $Srf_{f/f}$ (Parlakian et al., 2004) and conditional $Mrtfa^{-/-}Mrtfb^{f/f}$ mice (Mokalled et al., 2010) (see Maurice et al., 2024) after tissue dissociation with Liberase TM and TH (Roche, 43739557 and 43740136). Primary fibroblasts and SV40-immortalised MEFs were maintained in DMEM (Gibco, 41966-029) with 10% foetal calf serum (FCS; Gibco, 10270-106) and 1% Pen/Strep (Sigma, P4333). Where indicated, cells were starved for 5 days in 0.3% FCS. For all experiments with primary fibroblasts we included biological replicates, meaning cells isolated from three different mice for each genotype. For immortalised MEFs, cells were isolated from three individual embryos per genotype, yielding three WT MEF pools (79, 80 and 82) and three $Mrtfa^{-/-};Mrtfb^{fl/fl}$ MEF pools (84, 86 and 89) by Francesco Gualdrini (Treisman group), and three SRF WT (MEF1, MEF3 and MEF4) and three $Srf^{fl/fl}$ MEF pools [Bis1, 2+, and Bis3 derived from

$Srf^{tm1Zli};Gt(Rosa)26Sor^{tm9(cre/ESR1)Arte}$] were generated by Cyril Esnault (Treisman group). All immortalised and stable cell lines created in this study were routinely tested for mycoplasma and are available from the authors upon request. For all cell types (MEFs, primary fibroblasts and epithelial progenitor cells), $Mrtfb$ inactivation was induced by addition of 1 μM 4-hydroxytamoxifen (4OHT). For growth curves, 10,000 cells were seeded. Cells were treated with 5 μM H1152 or 10 μM blebbistatin as indicated.

### Retroviral transduction
Cell lines expressing Dox-inducible genes were constructed by lentiviral transduction of the $Mrtfa^{-/-}Mrtfb^{fl/fl}$ MEF pool 86. Lentiviral plasmids used were derivatives of pCW-MRTF-A-HA (Addgene plasmid 247436), which encodes doxycycline-inducible C-terminally HA-tagged full-length mouse MRTF-A, together with IRES-expressed GFP, derived from pCW (Addgene plasmid 41393). For inducible expression of RhoA$^{G14V}$ (Addgene plasmid 247437), CDK1 (Addgene plasmid 247438), CKS2 (Addgene plasmid 247439) or myoferlin (Addgene plasmid 247440), the MRTF-A sequences were replaced with appropriate cDNA. For lentivirus production, Phoenix cells (CRL-3214, ATCC) were transfected with pCW derivatives, psPAX2 (packaging; Addgene plasmid 12260), and VSVG (envelope; pMD2.G, Addgene plasmid 12259) in ratio 10:5:1 using Fugene HD (E2311, Promega) in OptiMEM (Gibco) with a Fugene:DNA ratio of 7:1, and selected using 1.25 μg/ml puromycin. Cells were used as pools, except for $Mrtfa^{-/-}Mrtfb^{fl/fl}$ DoxMRTF-A-HA cells, for which three cloned lines (1F2, 1F5, 1B2) were analysed. Protein expression was induced with 1 μg/ml doxycycline.

### CRISPR-Cas9 editing
Dox-inducible Cas9 was inserted into the mH11 locus of $Mrtfa^{-/-};Mrtfb^{fl/fl}$ MEF pool 86 using targeted CRISPR (SH100+SH305 targeting mH11, GeneCopoeia). Following selection in 2 μg/ml blasticidin, cloned cells were assayed for Dox-specific Cas9 editing, and line 2H17 was selected on the basis of growth behaviour comparable to the parental pool 86 cells (proliferation, SA-βGal, cell cycle protein expression, SASP mRNA expression, with and without 4OHT). For gene inactivation, 2H17 cells were infected with lentivirus expressing p27 single guide RNA (sgRNA) (Table S3) and selected with 1.25 μg/ml puromycin. Cas9 expression was induced with 1 μg/ml doxycycline, and three clonal lines – 3A2, 1F3 and 3B10 – of p27$^{-/-}$ $Mrtfa^{-/-};Mrtfb^{fl/fl}$ MEFs were isolated. Inactivation of p27 was confirmed by immunoblotting and sequencing (knockout scores >90%; Synthego, ICE analysis).

p27$^{-/-}$ $Mrtfa^{-/-};Mrtfb^{fl/fl}$ line 3A2 was then used to inactivate p21 and p57 using synthetic sgRNAs (Table S3), using RNAiMAX (13778150, Invitrogen) for sgRNA delivery. Following induction of Cas9 expression, three lines of p27$^{-/-}$ p21$^{-/-}$ p57$^{-/-}$ $Mrtfa^{-/-};Mrtfb^{fl/fl}$ MEFs (3.1D10, 3.1C11, 3.1G9) were cloned, and p21 and p57 inactivation confirmed by sequencing and immunoblotting. (knockout scores >90% and >80%; Synthego, ICE analysis). For p57, Sanger sequencing clone 3.1D10 revealed a 2 bp deletion in exon 2 (5′-CGAGACGG-3′ to 5′-CGA/CGG-3′).

### Epithelial cell cultures
Tracheal epithelial cultures were generated as described previously (Crotta et al., 2023; You et al., 2002). A total of 4–6 tracheas from 10–20-week-old female mice were dissected, pooled, cut in pieces and digested overnight at 4°C with 3 mg/ml Pronase (Roche, 11459643001) in MTEC basic medium [15 mM HEPES (Gibco, 83264), 0.03% NaHCO$_3$ (Gibco, 25080), 100 U/ml penicillin, 0.1 mg/ml streptomycin (Sigma, P4333), 2 mM L-glutamine (Gibco, 25030081) in DMEM/F12 medium (Gibco, 21331020)]. Cells (without cartilage) were washed with MTEC basic medium and treated with 0.2 mg/ml DNase (Sigma, D4527) in MTEC basic medium for 10 min at room temperature. Cells were washed with MTEC basic medium and resuspended in MTEC plus [MTEC basic medium containing 25 ng/ml EGF (BD, 354001), 0.1 mg/ml D-valine (Sigma, V1255), 30 μg/ml bovine pituitary extract (Gibco, 13028-014), 0.1 μg/ml cholera toxin (Sigma, C8052), 1× ITS (insulin, transferrin, selenium; Gibco, 41400045), 0.01 μM retinoic acid (Sigma, R2625), 10 μM Y27632 (Sigma, Y0503), 250 ng/ml amphotericin B (Gibco, A2942), 10% FCS]. After plating on an uncoated flask for 4 h to allow fibroblasts to adhere, the

epithelial cell supernatant was plated on 0.4 µg/cm² fibronectin (BD, 356008), 4 µg/cm² collagen (BD, 354236), treated with 4OHT the next day and cultured for 7 days until confluent. Cells were trypsinised (0.05% trypsin, 30 min; Gibco, 25300), resuspended in MTEC plus medium without Y27632 and seeded in non-differentiating conditions at 10,000 cells per insert on clear polyester membrane inserts (0.4 µm pores; Sarstedt, 833932041) coated with 0.4 µg/cm² fibronectin and 4 µg/cm² collagen in a 24-well plate.

### Adipocyte differentiation

MEFs were cultured for 20 days in DMEM (with 10% FCS and 1% Pen/Strep) containing 0.5 mM IBMX (I5879, Merck), 1 µM dexamethasone (D4902, Merck) and 10 µM insulin (I1882, Merck) without passaging. Cells were fixed in 10% neutral buffered formalin (NBF; Serosep, HistoPot), incubated in 60% isopropanol, stained with 1.8 mg/ml Oil Red O, 66% isopropanol, and rinsed in water. Imaging used a Zeiss Axio Observer Z1 inverted microscope equipped with an EC Plan-Neofluar 10×/0.30 objective and a QImaging colour camera.

### Immunoblotting

Cells were lysed in RIPA buffer [20 mM Tris-HCl pH 7.4, 150 mM NaCl, 0.1% sodium dodecyl sulfate (SDS), 0.5% Na-deoxycholate, 1% Triton X-100, 1× complete EDTA-free protease inhibitor cocktail (Roche, 54925800), 5 mM sodium fluoride and 1 mM sodium orthovanadate] and after spinning down cell debris, protein concentration was normalised using a Bradford assay (Bio-Rad, 5000006). After addition of Laemmli sample buffer, samples were run at 120 V on a NuPAGE 4–12% Bis-Tris gel (Invitrogen, NP0321BOX) in MES running buffer (Invitrogen, NP0002) and transferred to a nitrocellulose membrane (Amersham, 10600003) in transfer buffer (10% methanol, 192 mM glycine, 25 mM Trizma Base) at 200 mA for 1.5 h. Membranes were incubated in primary antibodies (Table S4) overnight followed by secondary antibodies (IRDyes, LICOR; Table S4) and developed using an Odyssey CLx (LICOR). Quantitative experiments were conducted with the three WT and three *Mrtfab*⁻/⁻ MEF pools. Marker expression level in each *Mrtfab*⁻/⁻ pool was determined relative to its internal tubulin control and expressed relative to the mean of its expression level in the three WT pools. Expression levels for each experiment are the mean of the three *Mrtfab*⁻/⁻ pools±s.e.m.

### Immunofluorescence and microscopy

Antibodies used are listed in Table S4. Cells were seeded on coverslips, fixed in 4% paraformaldehyde (PFA; Thermo Fisher Scientific, J19943.K2), permeabilised in 0.01% Triton X-100, blocked in 3% bovine serum albumin (BSA) and stained with primary antibodies at 4°C overnight. Coverslips were incubated with secondary antibodies in combination with DAPI and phalloidin–TRITC/647nm (Sigma, P1951 and Invitrogen, A22287), where indicated, and finally mounted in Mowiol mounting medium [9.32% (w/v) Mowiol 4-88, Calbiochem 475904]. Images were acquired with the ZEN 2.3 SP1software using a Zeiss LSM 710 confocal microscope equipped with either a Plan-Apochromat 40× or Plan-Apochromat 63× objective and an AxioCam camera or a Yokogawa CSU-W1 SoRa Spinning Disk system with a Nikon Ti2 inverted microscope equipped with a water immersion 40×/1.15 NA objective and a BSI express camera (Teledyne Photometrics) using the NIS-Elements software (Nikon Instruments, Inc.). Images were processed using ImageJ (NIH, Bethesda, MD, USA).

For adhesion and spreading analysis, cells were incubated with CellMask-Orange (Thermo Fisher Scientific, C10045) 1:1000 in growth medium for 10 min followed by fixation and mounting as above. Imaging used a Zeiss Microscope AXIO Observer.D1 equipped with a Plan-Neofluar 10× objective and an AxioCam MRm camera, using the ZEN 2012 (blue edition) software. Area and circularity were measured using ImageJ.

For Phasefocus Livecyte microscopy, cells were sparsely seeded in 12- or 24-well black glass-bottom plates. Images were acquired using the 10× objective and three large scan regions per well. Analysis used the Phasefocus analysis software.

### F-G actin fractionation

Cells were lysed in F-actin stabilisation buffer (100 mM PIPES pH 6.9, 5 mM MgCl₂, 1 mM EGTA, 30% glycerol, 0.1% Triton x-100, 0.1% NP-40, 0.1% Tween, 0.15% β-mercaptoethanol, 1 mM ATP and 1× EDTA-free protease inhibitors) and homogenised using a 25 G needle at 37°C. F- and G-actin fractions were obtained by ultracentrifugation at 100,000 *g* for 1 h at 37°C.

### Cell cycle phase, DNA synthesis and SA-βGal assays

Cells were pulsed with BrdU (19-160, Sigma-Aldrich, 10 µM) for 2 h, fixed in 100% ethanol on ice and treated with 2 M HCl for 30 min. Cells were stained using anti-BrdU antibody (BD Biosciences, 347580) and resuspended in 50 µl 100 µg/ml ribonuclease A (Sigma, R5125) and 150 µl 50 µg/ml propidium iodide (PI; Sigma, P4170) before analysis by flow cytometry.

For EdU incorporation assays, cells were pulsed for 6 h with Click-iT EdU with Alexa Fluor 488 (C10337, Invitrogen), fixed in 3.7% PFA and permeabilised in 0.5% Triton X-100, and incubated in the Click-iT reaction cocktail and imaged after Hoechst 33342 staining and mounting in Mowiol. Coverslips with micropatterns and micropillars were imaged using an Evident/Olympus VS200 slide scanner. Images were analysed in QuPath; EdU-positive cells were quantified using a threshold defined from a no EdU control, and Hoechst staining for total cell number.

SA-βGal activity analysis used the Senescence Cells Histochemical Staining Kit (Sigma, CS0030) and was imaged on a Zeiss Axio Observer Z1 inverted microscope equipped with an EC Plan-Neofluar 10×/0.30 objective and QImaging colour camera. For flow-cytometric analysis of SA-βGal, cells were fixed in 2% PFA, washed in 1% BSA in PBS, and stained overnight with CellEvent Senescence Green (Thermo Fisher Scientific, C10841) according to the manufacturer's protocol before analysis by flow cytometry.

For induction of proliferation phenotypes in WT cells by co-culture, primary kidney fibroblasts were infected with lentiviral particles containing pLVX–mCherry (Addgene plasmid 180646) 2 days after isolation, with 1.25 µg/ml puromycin selection for 5 days. Cells were then seeded either alone or in 100:1 co-culture with WT or *Mrtfab*-null MEFs for 5 days before CellEvent Senescence Green staining and analysis by flow cytometry.

Apoptosis was measured using a luminogenic caspase-3/7 substrate from the ApoTox-Glo Triplex Assay (G6320, Promega).

### RT-qPCR

RNA was extracted using GenElute Mammalian Total RNA miniprep kit (Sigma, RTN350-1KT) and cDNA synthesised using the Transcriptor First Strand cDNA Synthesis System (Roche, 0489703001). The cDNA was loaded into a MicroAmp Optical 384 well plate (Applied Biosystems, 4309849) and qPCR was carried out using 2× PowerUp SYBR Green Master Mix (Applied Biosystems, A25742). Absolute quantification of cDNA abundance was calculated using a mouse genomic DNA standard curve and then normalised to *Gapdh* or *Rps16* abundance. Datapoints are means of triplicate determinations. Primers used for RT-qPCR are listed in Table S3.

### RNA-seq

RNA was isolated as above and libraries were prepared using PolyA KAPA mRNA HyperPrep kit (Roche, KK8581) and sequenced with single-end read mode on a HiSeq Illumina platform. The analysis was carried out with nf-core/rnaseq v3.5 (https://nf-co.re/rnaseq/3.5/) using the GRCm38 as reference genome. Subsequent analysis was carried out in R version 4.4.3. Differential expression analysis was carried out with DEseq2 (Love et al., 2014). Following shrinking by the "ashr" method, log₂ fold change was plotted against the −log10 *P*-value to generate volcano plots. *P*-values were adjusted using the false discovery rate correction for multiple testing, and Gene Ontology and Gene Set Enrichment Analysis (GSEA) of the Hallmark category was performed (clusterprofiler; Xu et al., 2024). RNAseq data are available on GEO, accession number GSE298922.

### Adhesion assay

96-well plates were coated with ECM proteins [fibronectin, 40 µg/ml (Sigma, F2006); vitronectin, 20 µg/ml (Sigma, SRP3186); poly-L-lysine 10 µg/ml (Sigma, P4707); BSA 3%] diluted in PBS for 2 h at room temperature, then blocked with 3% filtered BSA in PBS for 15 min. 50,000 cells were seeded per well. At indicated timepoints, wells were washed in PBS and fixed in Crystal Violet solution (0.1% Crystal Violet, 20% methanol

in water). Plates were incubated at 4°C overnight, rinsed in milliQ water, and incubated on shaker in the dark with 100 μl 0.1% Triton X-100 per well. Absorbance was measured at 595 nm using a SpectraMax plate reader.

## Traction force measurements
Cells were seeded 12 days after 4OHT treatment on fibronectin-coated 30 kPa polyacrylamide gels containing fluorescent nanobeads (Invitrogen). Single cells were imaged using the Nikon CSU-W1 inverted microscope, brightfield images being acquired simultaneously with nanobead fluorescence images using a 40× objective. Cells were then trypsinised, and fluorescence images of beads were reacquired to record their position in the relaxed state. The gel deformation caused by the cells was analysed by comparing the bead positions with and without cells using a previously described Fourier transform algorithm (Butler et al., 2002; Trepat et al., 2009). The average force per unit area exerted by each cell was then calculated.

## Polyacrylamide gels for cell culture
Polyacrylamide gels of varying stiffness (Elosegui-Artola et al., 2016) were cast on a Bind Silane [1:14 Bind Silane solution (Sigma, M6514), 1:14 acetic glacial acid made up in 96% ethanol]-treated glass surface, using a Repel Silane (Sigma, GE17-1332-01)-treated coverslip to ensure a flat and even surface. Gel stiffness was adjusted by varying concentrations of 40% acrylamide and 2% bis-acrylamide in a PBS solution with 1:200 10% APS and 1:2000 TEMED. Gels were crosslinked in 5 mg/ml Sulpho-SANPAH (Sigma, 803332) in DMSO under UV for 10 min, and coated with 50 μg/ml fibronectin (Sigma, F2006) at 4°C overnight.

## Micropillars and micropatterns
For micropillars, a silicon wafer was spin-coated with SU-8 2005 photoresist (Y111045, Kayaku) at 1500 rpm on a Spin 150 spincoater (SPS-international) to achieve ~6 μm tall features. Soft baking was at 65°C for 1 min, then 95°C for 3 min. Photolithography was done at 120 mJ/cm² using an ML3 Microwriter to expose a pillar pattern of 1 μm diameter, 3 μm spacing, designed in Clewin. Post-exposure baking was at 65°C for 1 min, 95°C for 3 min, and 65°C for 1 min. The wafer was developed in propylene glycol methyl ether acetate (PGMEA) for 40 min, rinsed in isopropanol, dried with $N_2$ and hard baked at 200°C for 20 min. Polydimethylsiloxane (PDMS; SYLGARD 184, 10:1 base: curing agent) was applied to the patterned wafer and spin-coated under vacuum to form a thin layer. After degassing, the PDMS was cured at 110°C for 5 min, peeled off, and placed (pillars up) onto plasma-cleaned coverslips. A final bake at 110°C for 15 min was performed. Pillars were coated with 50 μg/ml fibronectin overnight.

For micropatterning, coverslips were cleaned in 1 M HCl, plasma treated, and incubated with near-IR labelled PLL-PEG [PLL(20)-g[3.5]-PEG(2)/Atto633; SuSoS]. Micropatterns were generated in the PEG layer using a 185 nm UV lamp (UVO 342-220, Jelight) with a custom-made quartz photomask containing the desired patterns (Compugraphics). The UV-exposed surface was coated overnight with 50 μg/ml fibronectin in $NaHCO_3$ prior to cell seeding.

## OPP incorporation for protein biosynthesis
Protein biosynthesis was measured using Click-&-Go Plus OPP Protein Synthesis Assay Kit (Click Chemistry Tools, 1493). Cells were pulsed with 20 μM O-propargyl-puromycin (OPP) for 30 min, fixed in 4% PFA and permeabilised in 0.5% Triton X-100. After the OPP–Alexa Fluor 488 click reaction, nuclei were stained with Hoechst 33342. Coverslips were mounted in Mowiol and imaged using a Zeiss Microscope AXIO Observer.D1 equipped with a Plan-Neofluar 10× objective and an AxioCam MRm camera. In ImageJ, cells were segmented using the nuclear stain for marker-controlled watershed segmentation, and the mean OPP intensity per cell was measured.

## Acknowledgements
We thank Patrick Costello, lab members, Michael Way and Francesco Gualdrini for helpful discussions, and Gerard Evan, Michael Way and Paul Nurse for helpful comments on the manuscript. We thank Cyril Esnault and Francesco Gualdrini for the immortalised MEF cell pools, Stefania Crotta and Andreas Wack for help with tracheal epithelial cell culture, the Way lab for pLVX-mCherry plasmid (Addgene #180646). Mike Howell and Ming Jiang from the Crick High-throughput screening STP for help with generation of the Cas9 cell line, Robert Goldstone and Deb Jackson in the Crick Advanced sequencing STP for library preparation and sequencing, and Donald Bell and Matthew Renshaw from the Crick Advanced Light Microscopy STP for guidance and assistance using microscopes and slide scanner. This work was supported by the Francis Crick Institute which receives its core funding from Cancer Research UK (CC2102), the UK Medical Research Council (CC2102), and the Wellcome Trust (CC2102).

## Competing interests
The authors declare no competing or financial interests.

## Author contributions
Conceptualization: J.C.N., S.B., R.T.; Data curation: J.C.N.; Formal analysis: J.C.N.; Funding acquisition: R.T.; Investigation: J.C.N., M.B.-J., N.C.P., J.D., S.B.; Supervision: J.C.N., R.T.; Writing – original draft: J.C.N., N.C.P., R.T.; Writing – review & editing: J.C.N., M.B.-J., N.C.P., J.D., S.B., R.T.

## Funding
This work was supported by the Francis Crick Institute, which receives its core funding from Cancer Research UK (CC2102), the UK Medical Research Council (CC2102) and the Wellcome Trust (CC2102). This research was funded in whole, or in part, by the Wellcome Trust CC2102. M.B-J. was recipient of an H2020 individual fellowship H2020-MSCA-IF (897131). Open Access funding provided by the Francis Crick Institute. Deposited in PMC for immediate release.

## Data and resource availability
RNAseq data are available on NCBI GEO, accession number GSE298922. All other relevant data and details of resources can be found within the article and its supplementary information.

## Peer review history
The peer review history is available online at https://journals.biologists.com/jcs/lookup/doi/10.1242/jcs.264444.reviewer-comments.pdf

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
