## [Peer Review File · Journal of Cell Science]

MRTF-dependent cytoskeletal dynamics drive efficient cell cycle progression

Julie C. Nielsen, Maria Benito-Jardon, Noel Christo Petrela, Jessica Diring, Sofie Bellamy and Richard Treisman
DOI: 10.1242/jcs.264444

Editor: Renata Basto

Review timeline

Submission to Review Commons:	8 June 2025
Submission to Journal of Cell Science:	12 September 2025
Accepted:	21 September 2025

Reviewer 1

Evidence, reproducibility and clarity

The manuscript by Neilsen et al. presents a thorough and well-structured study showing that Myocardin-related transcription factors (MRTF-A/B), via MRTF-SRF, are essential for the proliferation of both primary and immortalized fibroblasts and epithelial cells. Using a combination of knockouts/rescue experiments, cytoskeletal analysis, and transcriptomics, the authors demonstrate that MRTF-SRF signalling controls actin dynamics and contractility- key drivers of cell cycle progression. Notably, they show that the proliferative arrest caused by MRTF loss is reversible, distinguishing it from classical senescence.

Major points

1. The link between MRTF-SRF activity, cytoskeletal organisation, and cell proliferation is clearly established. The fact that disrupting contractility phenocopies MRTF loss strengthens the case that the pathway acts through mechanical control.
2. The authors support their conclusions using multiple cell types (MEFs, primary fibroblasts, epithelial cells), a range of complementary assays (RNA-seq, traction force microscopy, adhesion/spreading), and genetic tools (CRISPR, inducible rescue).
3. The ability to restore proliferation by re-expressing MRTF-A argues against true senescence and instead suggests a quiescence-like state driven by cytoskeletal disruption.
4. This work particularly highlights how mechanical inputs feed into transcriptional programs to regulate proliferation, with implications for understanding anchorage-dependent growth.

Suggestions

1. While the authors argue convincingly against classical senescence, elevated SA- β Gal and SASP expression suggest a more nuanced arrest state. It not really clear what this state is or is not, therefore a deeper discussion of possible hybrid or intermediate states would be helpful - maybe potential additional experiments to include or exclude potential explanations
- e.g. how does it differ from G0 exit?
2. The transcriptomic data are strong, but the paper would benefit from zooming in on specific MRTF-SRF targets (e.g., actin isoforms, adhesion molecules) that directly link cytoskeletal regulation to cell cycle control.
3. It depends on where what target journal would be, but this is a very well executed mechanistic study that doesn't really have an impact. Extending the discussion to human systems- or tissues where contractility is critical-could broaden the impact and applicability of the findings.

4. As above, the paper briefly mentions transformation, but it would be valuable to elaborate on whether MRTF-SRF acts as a barrier or enabler in tumorigenesis under different conditions. This I feel is the main weakness remaining - e.g. it would be fine with enabling different effects driven by other transcription events in emerging tumour cells (oncogenic in context of RAS, suppressive in context of p53) but I think the manuscript fails to be definitive on this points. Addressing this would make a much stronger and impactful study. I believe they have an impact peice of science that outlines how mechanical events impact cell fate decisions, but this is unlikely to be the driver - ie it facilitates cell fate decisions in context of tissue stiffness.

Significance

Overall

This is a well-executed and insightful study that deepens our understanding of how cytoskeletal signals drive proliferation through MRTF-SRF. It broadens the role of this pathway beyond motility and offers new perspectives on mechanotransduction and cellular plasticity. If is weak in its demonstration of biological significance, but if the aim to to present a pure basic cell biology story it is good.

Reviewer 2

Evidence, reproducibility and clarity

In this manuscript, Nielsen and colleagues examine the impact of MRTF-A/B and SRF gene inactivation on cell proliferation. They performed an extensive body of work (using multiple cell types and multiple clones) to show that MRTF inactivation causes cell cycle arrest and senescence (mimicking the phenotype of SRF knockout cells) although the changes in the expression of various CDK inhibitors were cell-type specific. Very interestingly, simultaneous inactivation of all three major CDK inhibitors failed to rescue MRTF knockout cells from their proliferation defect. Expectedly, MRTF knockout cells exhibited defects in actin cytoskeleton, adhesion, and contractility. Interestingly, hyperactivating Rho also failed to rescue MRTF knockout cells from proliferation defect. The main conclusion of the paper was derived from experiments which showed that inhibition of either ROCK or myosin caused wild-type cells to behave like MRTF knockout cells rather than demonstration of any molecular perturbation that could reverse the proliferation defect of MRTF knockout cells. While the experimental studies are thorough and rigorous, a vast majority of the core findings related to the loss-of- function of MRTF that are reported herein (i.e. defects in cell proliferation, elevation of CDK inhibitors, migration, actin cytoskeleton, contractility) are not conceptually new and have been previously reported in other cell systems by several investigators including this research group. In the reviewer's opinion, since the authors have not been able to identify a molecular strategy to reverse the proliferation phenotype of MRTF knockout cells, the underlying mechanisms of MRTF-dependent regulation of cell proliferation remain largely unanswered.

Other comments: Majority of the immunoblot data have not been quantified. P16 data in Fig 1G vs Fig S1A are not similar (although the authors mention that the findings are similar)

Significance

This study aims to investigate a fundamental biological question of how an actin-regulated transcription machinery regulates cell proliferation and is therefore of broad significance. Strengths and limitations of this study are described above.

Reviewer 2

Evidence, reproducibility and clarity

Summary

The manuscript by Nielsen et al. (Treisman lab) entitled "MRTF-dependent cytoskeletal dynamics drive efficient cell cycle progression" investigates the effects on cell proliferation elicited upon cellular depletion of the transcription factors MRTF-A and MRTF-B. The MRTFs are actin-dependent co-factors of SRF, which direct the transcription of SRF target genes. The MRTF-SRF regulatory circuit defines both the functioning and the control of actin-driven cytoskeletal dynamics. The work presented identifies essential molecular links that interconnect cytoskeleton-dependent cellular activities (cell-cell adhesion, cell-substrate contact, cell spreading) and cell proliferation.

General assessment on used methodology.

The presented comprehensive body of work is performed competently; it includes all relevant and necessary state-of-the-art technologies.

Significance

Advance

Previously published evidence by others (including the Treisman group) had indicated that SRF does not seem essential for the proliferation of some cell types (i. e., embryonic (stem) cells, activation-dependent immune cells, etc.). In regard to this, the authors discuss in the current manuscript: "Although further work is needed to elucidate the basis for these context-dependent differences, our data show that MRTFSRF signalling is likely to play a more general role in proliferation than previously thought." The current manuscript already delineates this "general role": MRTF-SRF signalling impinges on cell proliferation whenever proliferative activities are dependent upon cytoskeletal dynamics.

The work has implications for cancer biology. It offers new directions to investigate the regulation of proliferative activities of anchorage-independent tumor cells.

Audience

The insights generated serve the wide interests of a large and diverse group of cell and tumor biologists.

Reviewers field of expertise (keywords). Cytoskeletal dynamics, transcriptional control.

Author response to reviewers' comments

Manuscript number: RC-2025-03062

Corresponding author(s): Treisman, Richard

1. General Statements [optional]

We thank the three reviewers for their comments on the paper.

We are pleased to see that they consider it be a comprehensive and well-executed study, which clearly establishes a previously overlooked connection between MRTF- SRF signalling and proliferation, and that its conclusions require no further experimentation.

As review 3 points out, this work has implications for cancer biology, and suggests new research routes to understand the relation between cell adhesion, proliferation, and transformation.

However, two referees raise significant concerns about its impact

Review 1 suggests that the paper lacks impact without exploration the wider biological significance of our observations, although it considers it to be a good basic cell biology study. It suggests further work extending the findings to tissue- or tumor-based systems. While we consider such

studies worthwhile - indeed we are currently pursuing these directions - we consider them beyond the scope of the present paper.

Review 2 questions the novelty of our findings. We strongly disagree. This is the first study to show that MRTF-SRF signalling is required for the proliferation of both primary and immortalised fibroblasts, and epithelial cells. We show that MRTF inactivation leads cells to enter a quiescence-like state under conditions that would permit efficient cell cycle progression in wildtype cells. The study will alter the field's perspective on the role of MRTF-SRF signalling, previously viewed as concerned with cell adhesion, morphology, and motility.

Responses to individual reviews follow (in blue text).

This section is mandatory. Please insert a point-by-point reply describing the revisions that were already carried out and included in the transferred manuscript.
(author response to reviewers in blue, specific changes made to text in bold)

Reviewer #1

(Evidence, reproducibility and clarity (Required)):

The manuscript by Neilsen et al. presents a thorough and well-structured study showing that Myocardin-related transcription factors (MRTF-A/B), via MRTF-SRF, are essential for the proliferation of both primary and immortalized fibroblasts and epithelial cells. Using a combination of knockouts/rescue experiments, cytoskeletal analysis, and transcriptomics, the authors demonstrate that MRTF-SRF signalling controls actin dynamics and contractility-key drivers of cell cycle progression. Notably, they show that the proliferative arrest caused by MRTF loss is reversible, distinguishing it from classical senescence.

Major points

1. The link between MRTF-SRF activity, cytoskeletal organisation, and cell proliferation is clearly established. The fact that disrupting contractility phenocopies MRTF loss strengthens the case that the pathway acts through mechanical control.
2. The authors support their conclusions using multiple cell types (MEFs, primary fibroblasts, epithelial cells), a range of complementary assays (RNA-seq, traction force microscopy, adhesion/spreading), and genetic tools (CRISPR, inducible rescue).
3. The ability to restore proliferation by re-expressing MRTF-A argues against true senescence and instead suggests a quiescence-like state driven by cytoskeletal disruption.
4. This work particularly highlights how mechanical inputs feed into transcriptional programs to regulate proliferation, with implications for understanding anchorage-dependent growth.

Suggestions

1. While the authors argue convincingly against classical senescence, elevated SA- β Gal and SASP expression suggest a more nuanced arrest state. It not really clear what this state is or is not, therefore a deeper discussion of possible hybrid or intermediate states would be helpful - maybe potential additional experiments to include or exclude potential explanations - e.g. how does it differ from G0 exit?

Our findings show that MRTF inactivation inhibits cell proliferation under conditions that would permit efficient cell cycle progression in wildtype cells, inducing a state with some features associated with classical senescence, and others conventionally associated with reversible cell cycle arrest/quiescence. The reviewer correctly points out that this raises problems with accurately defining the nature of the MRTF-null proliferation defect.

To our knowledge there are no rigorously defined unambiguous markers for senescence,

quiescence, or G0. Indeed, recent studies have shown that senescence and quiescence / G0 states are not as distinct as previously assumed (Anwar et al, 2018; Ashraf et al 2023) as we reviewed in detail in Discussion p27, §2; p28 §3. We therefore do not consider it a productive endeavour to define markers for the MRTF-null state as opposed to defining its mechanistic basis. However, we agree that we should have been clearer about how the phenotypes we observe relate to classical cell arrest states.

We have therefore revised the presentation of the Results to make it clear which features of the non-proliferative state associated with MRTF inactivation are seen in classical senescence, and which are found in reversible cell cycle exit or quiescence.

Things done:

- **Results pp16-17 and Fig 1. Figure panels and presentation are reordered to present “senescence” features together before marker expression (panel G is now panel I). Text now explicitly points out that the spectrum of cell cycle markers, specifically p27 upregulation, is not that associated with classical senescence (p16, p21, etc) but previously linked to reversible arrest or quiescence. Lines 371-380 have been moved up from the succeeding paragraph; statement added *re p27* and reversible cell cycle exit on lines 387-389; summary sentence added in lines 398-401).**
- **Statement added that reversibility distinguishes the MRTF defect from classical senescence p20§1 line 454-455.**
- **Note that p27 is associated with reversible arrest included on p20§2 line 460.**

We also explicitly summarised the features of the phenotype at the start of the Discussion.

- **Sentences added p27§1 lines 626-631.**
- **Emphasis that p27 protein upregulation is associated with reversible cell cycle inhibition and quiescence is added on p28 lines 668-669.**

2. The transcriptomic data are strong, but the paper would benefit from zooming in on specific MRTF-SRF targets (e.g., actin isoforms, adhesion molecules) that directly link cytoskeletal regulation to cell cycle control.

We have now clarified presentation of the RNAseq data in Figure 5 and the data summary tables. Figure 5B now identifies which of those genes showing deficits in MRTF-null MEFs were previously identified as direct genomic targets for MRTF-SRF, and that the majority are cytoskeletal.

- **Additional columns added in Table 1 to indicate whether genes are candidate genomic MRTF-SRF targets; Table 2 now show gene symbol lists as well as ENSEMBL IDs for GO categories and NCBI Entrez IDs for GSEA categories, respectively.**
- **Figure 5B revised to point out cytoskeletal genes that are genomic MRTF- SRF targets in bold, legend clarified p40 lines 920-922.**
- **Now noted p23 lines 527-529 that cytoskeletal genes affected include many direct MRTF-SRF targets.**

Our data confirms that in MEFs, MRTF inactivation affects fibroblast cell morphology, adhesion, spreading, motility and contractility (Figures 5, 6), as seen in many other settings.

A critical question remains as to whether these effects reflect limitation in one MRTF target gene or several, and how this defect relates to proliferation.

Concerning specific MRTF-SRF gene targets:

Cells lacking cytoplasmic actins are reported to exhibit defective proliferation, (**now noted in Results p23 lines 529-532**). We are currently evaluating whether this defect has similarities with the MRTF-null proliferation phenotype (see Discussion p31, §2).

Previous findings suggest that defective cytoplasmic actin expression may underlie most MRTF knockout phenotypes (Salvany *et al*, 2014; Maurice *et al.*, 2024) previously noted in the Discussion (see p31, §2).

The myoferlin gene promotes growth of liver cancer cells by inhibiting ERK activation and oncogene induced senescence. We showed that myoferlin expression does not promote proliferation of MRTF-null MEFs in the original submission (see Figure S5E). Additionally, we now point out that the RNAseq data show that myoferlin expression is not significantly affected in MRTF-null MEFs (**new text p23, lines 532-534**).

3. It depends on where what target journal would be, but this is a very well executed mechanistic study that doesn't really have an impact. Extending the discussion to human systems or tissues where contractility is critical could broaden the impact and applicability of the findings.

We interpret this comment as indicating that our paper does not address the wider biological implications of our findings by extension to studies in tissue or tumour systems.

As outlined in our response to review 3, our study provides strong evidence that MRTF-SRF will be required for cell proliferation in settings where physical progression through cell cycle transitions requires high contractility, either owing to intrinsic factors or external physical constraints such as tissue stiffness, fibrosis, or tumour microenvironment.

Discussion now explicitly addresses potential roles for tissue stiffness (pp30§2 lines 717-718, and p32§1 725-727). However, we feel that resolution of this question is beyond the scope of the present paper.

4. As above, the paper briefly mentions transformation, but it would be valuable to elaborate on whether MRTF-SRF acts as a barrier or enabler in tumorigenesis under different conditions. This I feel is the main weakness remaining - e.g. it would be fine with enabling different effects driven by other transcription events in emerging tumour cells (oncogenic in context of RAS, suppressive in context of p53) but I think the manuscript fails to be definitive on these points. Addressing this would make a much stronger and impactful study. I believe they have an impact piece of science that outlines how mechanical events impact cell fate decisions, but this is unlikely to be the driver - ie it facilitates cell fate decisions in context of tissue stiffness.

We find it difficult to understand the precise points being made here.

However, transformation has long been known to bypass physical constraints on proliferation such as the requirement for adhesion. Moreover, MRTF-SRF activity is not necessarily required for proliferation of all transformed cells (Hampl *et al*, 2013; Medjkane *et al*, 2009; our unpublished data). The relation of our findings to transformation is thus an open question, which we are actively pursuing. **Now noted in revised Discussion p32, lines 752-755**.

MRTF-independent proliferation of tumor cells could reflect oncogenic signals substituting for MRTF-dependent ones (eg from focal adhesions), or from relief of cytoskeletal constraints on proliferation (adhesion independent proliferation). In contrast, in proliferation of DLC1-deleted cancer cells is dependent on suppression of oncogene-induced senescence by MRTF-SRF signalling (Hampl *et al*, 2013). These points were already made in Discussion p28, pp30-31.

Although our current work is focussed on cell transformation, we would respectfully suggest the in-depth resolution of this complex question is beyond the scope of the present paper.

See also response to (3) above.

Reviewer #1 (Significance (Required)):**Overall**

This is a well-executed and insightful study that deepens our understanding of how cytoskeletal signals drive proliferation through MRTF-SRF. It broadens the role of this pathway beyond motility and offers new perspectives on mechanotransduction and cellular plasticity. It is weak in its demonstration of biological significance, but if the aim is to present a pure basic cell biology story it is good.

The vast majority of work with the SRF system has led to the common perception that its role is exclusively with cell motility and adhesive processes, not proliferation. The results presented in the paper, even if limited to cell culture models, are therefore novel.

Reviewer #2

(Evidence, reproducibility and clarity (Required)):

In this manuscript, Nielsen and colleagues examine the impact of MRTF-A/B and SRF gene inactivation on cell proliferation. They performed an extensive body of work (using multiple cell types and multiple clones) to show that MRTF inactivation causes cell cycle arrest and senescence (mimicking the phenotype of SRF knockout cells) although the changes in the expression of various CDK inhibitors were cell-type specific.

Very interestingly, simultaneous inactivation of all three major CDK inhibitors failed to rescue MRTF knockout cells from their proliferation defect. Expectedly, MRTF knockout cells exhibited defects in actin cytoskeleton, adhesion, and contractility. Interestingly, hyperactivating Rho also failed to rescue MRTF knockout cells from proliferation defect. The main conclusion of the paper was derived from experiments which showed that inhibition of either ROCK or myosin caused wild-type cells to behave like MRTF knockout cells rather than demonstration of any molecular perturbation that could reverse the proliferation defect of MRTF knockout cells.

While the experimental studies are thorough and rigorous, a vast majority of the core findings related to the loss-of-function of MRTF that are reported herein (i.e. defects in cell proliferation, elevation of CDK inhibitors, migration, actin cytoskeleton, contractility) are not conceptually new and have been previously reported in other cell systems by several investigators including this research group.

This is the first study showing that MRTF-SRF signalling is required for the proliferation of both primary and immortalised fibroblasts, and epithelial cells. We show that the MRTF-SRF non-proliferative state combines features of both classical senescence and reversible cell cycle exit / quiescence.

The vast majority of previous work with the SRF system has led to the common perception that its role is exclusively related to cell motility and adhesive processes and not proliferation (see Olson and Nordheim 2010). Where proliferation has been examined directly, both others and our own previous studies of the MRTFs in immune cells and cancer cell lines have revealed no direct role in proliferation (Schratt et al, 2001; Medjkane et al 2009; Maurice et al, 2024). The results presented here are therefore novel.

In the reviewer's opinion, since the authors have not been able to identify a molecular strategy to reverse the proliferation phenotype of MRTF knockout cells, the underlying mechanisms of MRTF-dependent regulation of cell proliferation remain largely unanswered.

Indeed, our attempts to rescue the phenotype (knockouts of the CKIs, and overexpression of different downregulated factors) did not restore proliferation. We therefore now aim to attack the problem (i) through overexpression screens, and (ii) by identifying differences between

MRTF-SRF dependent and -independent (eg transformed) cells. However, these are new projects that are beyond the scope of a revised paper.

Other comments: Majority of the immunoblot data have not been quantified.

P16 data in Fig 1G vs Fig S1A are not similar (although the authors mention that the findings are similar)

We have addressed these issues by reorganisation and quantification the immunoblotting data as follows:

- Figure S1A has been moved to new Figure 1I, replacing the limited analysis shown in old Figure 1G. This more comprehensive, and displays data from all three WT and *Mrtfab*^{-/-} pools.
- Figure 1I data is quantified. Marker expression in each *Mrtfab*^{-/-} pool is evaluated relative its mean expression in the three WT pools treated in parallel.
- A new Figure S1A shows mean marker expression across the three *Mrtfab*^{-/-} pools, drawn from 5 independent analyses (not all markers included in each analysis). Different analyses of marker expression may exhibit variation, resulting from differences in handling, culture medium, plating density, relative confluence, etc. However, *Mrtfab*^{-/-} cells exhibit markedly increased p27 and TLR2 expression, while expression of the other markers tested, including p16, consistently decreases.
- Spearman comparisons among the WT and *Mrtfab*^{-/-} pools show that relative marker expression is indeed well correlated between the pools of each genotype.

Note on quantitation added in Methods p10 lines 209-213.

Figure 1I moved from former Figure S1A, to replace former Figure 1G. New legend now includes quantitation, and reference to Spearman correlations, p44 lines 834-841.

New Figure S1A displays data from multiple independent experiments with all 3 *Mrtfab*^{-/-} pools. New legend, p44 lines 997-1002.

Figure S1B legend notes correlation between relative marker expression in untreated WT and *Mrtfab*^{-/-} cells, p44, lines 1005-1008. Results text rewritten p17 lines 383-391; no reference to "similar".

Reviewer #2 (Significance (Required)):

This study aims to investigate a fundamental biological question of how an actin- regulated transcription machinery regulates cell proliferation and is therefore of broad significance. Strengths and limitations of this study are described above.

Reviewer #3

(Evidence, reproducibility and clarity (Required)):

Summary

The manuscript by Nielsen et al. (Treisman lab) entitled "MRTF-dependent cytoskeletal dynamics drive efficient cell cycle progression" investigates the effects on cell proliferation elicited upon cellular depletion of the transcription factors MRTF-A and MRTF-B. The MRTFs are actin-dependent co-factors of SRF, which direct the transcription of SRF target genes. The MRTF-SRF regulatory circuit defines both the functioning and the control of actin-driven cytoskeletal

dynamics.

The work presented identifies essential molecular links that interconnect cytoskeleton-dependent cellular activities (cell-cell adhesion, cell-substrate contact, cell spreading) and cell proliferation.

General assessment on used methodology.

The presented comprehensive body of work is performed competently; it includes all relevant and necessary state-of-the-art technologies.

Reviewer #3 (Significance (Required)):

Advance

Previously published evidence by others (including the Treisman group) had indicated that SRF does not seem essential for the proliferation of some cell types (i. e., embryonic (stem) cells, activation-dependent immune cells, etc.). In regard to this, the authors discuss in the current manuscript: "Although further work is needed to elucidate the basis for these context-dependent differences, our data show that MRTF-SRF signalling is likely to play a more general role in proliferation than previously thought." The current manuscript already delineates this "general role": MRTF-SRF signalling impinges on cell proliferation whenever proliferative activities are dependent upon cytoskeletal dynamics.

We of course support the view that it is MRTF-SRF's role in cytoskeletal dynamics, especially contractility, that is a limiting factor for cell cycle progression in our cells; however, this may not be the cases or other cell types or settings, such as adhesion-independent or transformed cells, and/or stiff tissue environments.

We have stated this view more strongly, modifying the abstract and discussion, and rewording the sentence quoted above.

The major point is that MRTF-SRF-dependent proliferation may be more common than previously thought, the field having focussed on its role in cytoskeletal dynamics rather than proliferation.

Abstract lines 48-49; Discussion p28, line 668-669; pp30-31, lines 713-714, 725-727. See also last para pp31/32, added lines 752-755.

The work has implications for cancer biology. It offers new directions to investigate the regulation of proliferative activities of anchorage-independent tumor cells.

Audience

The insights generated serve the wide interests of a large and diverse group of cell and tumor biologists.

Reviewers field of expertise (keywords).

Cytoskeletal dynamics, transcriptional control.

Original submission

First decision letter

MS ID#: jcs.264444

MS Title: MRTF-dependent cytoskeletal dynamics drive efficient cell cycle progression

Authors: Julie Charlotte Nielsen; Maria Benito-Jardon; Noel Christo Petrela; Jessica Diring; Sofie Bellamy; Richard Treisman

Article Type: Review Commons Transfer

Dear Dr Treisman,

I am happy to tell you that your manuscript has been accepted for publication in Journal of Cell Science, pending standard publication integrity checks.

Thank you for sending your manuscript to Journal of Cell Science through Review Commons.